

# Impacts of a lengthening open water season on Alaskan coastal communities

Rebecca J. Rolph[1], Andrew R. Mahoney[1], John Walsh[2], and Philip A. Loring[3]

[1]Geophysical Institute, University of Alaska Fairbanks, PO Box 757320, Fairbanks, Alaska 99775-0001
[2]International Arctic Research Center, Fairbanks, Alaska, USA
[3]School of Environment and Sustainability, University of Saskatchewan, Saskatoon, Saskatchewan, Canada

*Correspondence to:* Rebecca J. Rolph (rjrolph@alaska.edu)

**Abstract.** It is often remarked that Arctic coastal communities are on the frontlines of the impacts related to the rapidly diminishing ice pack. These impacts can have direct effects on communities, such as reduced access to subsistence hunting species, or increased wave height and coastal erosion. There are also indirect effects driven by external socioeconomic systems, such as increased maritime activity, which may provide local economic benefits while increasing potential for disruption to subsistence activities. Here, we use the Historical Sea Ice Atlas (HSIA) dataset to assess the potential direct and indirect impacts from sea ice change for selected Alaska communities. The HSIA provides sea ice concentration for the Bering, Chukchi and Beaufort Seas on a 0.25-degree grid for the period 1953-2013. We estimate the timing of freeze-up and break-up, which is reported by local residents to be of critical importance for subsistence hunting activities and food security. We calculate the open water season length and extend the existing timeseries of the Barnett Severity Index (BSI), which assesses the impact of ice conditions on maritime traffic destined for the Beaufort Sea. We find consistent trends toward later freeze-up and earlier break-up, leading to a lengthened open water period. In Utqiavik (formerly Barrow), there is evidence of a navigational regime change in the 1990s when the pack ice edge started to routinely retreat beyond this most northern community.

## 1  Introduction

Climate change is one of the multiple global drivers impacting the food security of northern communities (Loring and Gerlach, 2015), and the impacts of climate change can be separated into two distinct categories: direct and indirect impacts. The proposed definition of direct impacts here are those impacts directly induced by climate change that are not influenced by global socio-economic systems. Examples of such direct impacts include accelerated rates of erosion related to increased critical wave height along coastlines due to reduced sea ice (Overeem et al., 2011). Additionally, changes in the availability and trafficability of sea ice lead to changes in subsistence hunting practices (Laidler et al., 2009), reduced reliability of safe travelling conditions (Barber et al., 2014), while changes in the seasonality and extent changes migration of important subsistence animals (Moore and Huntington, 2008). These impacts have been well investigated and documented in various media (e.g. Gearheard (2013); Gearheard et al. (2006); Laidler (2006); Lovecraft and Eicken (2011)). By contrast, globally-induced impacts are those that are not a direct result of a changing physical environment, but ones that result from the interaction between the climate system



and global socio-economic systems. For example, a longer navigable ice season may result in increased barge traffic and oil exploration, but only if economic conditions allow.

The challenges associated with the place-based nature of climate change impacts have also been documented, such as how people respond to environmental change (Loring et al., 2016). Challenges also include appropriate definitions of what

constitutes acceptable impacts of research, and the associated extent and nature of local benefits that research projects should provide (Gearheard and Shirley, 2007). When trying to downscale large-scale climate observations, complex inter-connections between communities and the environment can often be overlooked (Huntington et al., 2009). We therefore recognize the importance of incorporating local knowledge in understanding and quantifying the impacts of such direct changes (Huntington et al., 2009).

Community interaction and feedback is useful to identify the social relevance of climate system variables commonly used by scientists. For example, sea ice concentration, thickness, and extent can be considered "primary" geophysical variables for an Arctic ocean study, but for indigenous Arctic coastal communities, timing of freeze-up and break-up are among the most important criteria that define the state of the sea ice regime locally (Berkes and Jolly, 2002; Berman and Kofinas, 2004; Laidler et al., 2009). Residents of Arctic coastal communities report that the sea ice is changing in many other ways, including

increased presence of rotten ice and the way the ice breaks up (Kirk, 2013). While these changes are unlikely to be directly captured in climate-scale observations, local community members connect these changes with trends that can be indicated from the length of the transition season. This type of metric can be examined using large-scale datasets. Thus, incorporation of indigenous knowledge allows the application of large-scale datasets to examine relevant local impacts.

In this study, we use the Historical Sea Ice Atlas (HSIA) dataset to quantify changes in sea ice in the Alaska Arctic over

the last several decades, and explore the idea that globally-induced effects on local communities due to climate change can be analyzed separately from those induced by direct climate change impacts.

Specific examples of Arctic rural communities are presented here to highlight the differences between the direct effects of climate change and the indirect global impacts. We explore past events and local community differences in the magnitude of the increasing open water window.

**2  Data and Methods**

The Historical Sea Ice Atlas extends back to 1850 with monthly resolution, incorporating various datasets from whaling ship logs taken at that time to historical ice chart archive products, as well as best analog representations of a given month filling any temporal or spatial gaps (Walsh et al., 2016; Scenarios Network for Alaska and Arctic Planning, 2015). From 1953 through 2013, quarter-monthly sea ice concentration values are available and were used here to investigate trends in changes in sea

ice concentration along the coastlines of selected communities in Alaska. The communities of Barrow, Kotzebue, Shishmaref, and Nome were selected to represent a range of sea ice regimes, with varying levels of dependence on subsistence activities, such as susceptibility to coastal erosion and interaction with the offshore oil and gas industry. In addition, an offshore area was



analyzed in the Bering Strait. Figure 1 shows the grid cells selected for each community and offshore areas used to analyze ice concentration.

Use of HSIA data roughly doubles the timespan we can consider as compared with relying only on satellite-derived sea ice data. The HSIA has a 0.25 x 0.25 degree resolution, and is a compilation of a variety of different sources of sea ice concentration data. A full list of all sources of data into the HSIA can be found on the Scenarios Network for Arctic and Alaska Planning (SNAP) webpage (http://seaiceatlas.snap.uaf.edu/about). While the quarter monthly data starts in 1953, monthly data in the HSIA goes back to 1850. One caveat to including multiple data sources is that there are a different the number of observations that have gone into each time segment of data. For example, the frequency of ship-based of observations was not consistent throughout the record and the number of available observations increased dramatically with the advent of the satellite era. However, with regard to the latter, we do not find evidence for anomalous discontinuity in trends around 1978-79, when the passive microwave satellite data are incorporated into the dataset. The HSIA includes a land-mask, so grid cells near the coastline can be readily identified.

Coastal areas are utilized by rural Alaska residents for subsistence hunting and fishing. Coastal areas are also used for transportation and barge deliveries. All of these are impacted by the presence or absence of sea ice (though we note that barge deliveries and other maritime traffic may be affected by non-local ice conditions as well, particularly at 'choke points' en route). To assess variability in ice conditions most immediately relevant to the security of coastal communities (Loring et al., 2013), we selected data from within 50 km of the coastline and 75 km along the coastline from each community. The maximum concentration (greatest fraction of sea ice per unit area) was extracted from within this area. For the offshore area analyzed (Bering Strait), the maximum ice concentration from a 0.5 degrees longitude and 0.75 degrees latitude box was examined. Quarter-monthly data provided by the HSIA were assigned a calendar date by the best approximation of the midpoint day of each quarter-monthly file. The open water period is defined as the duration between the assigned calendar days in the summer season where the ice concentration stayed below 30%. Freeze-up day and break-up day are defined as the time when the ice passes the thirty percent concentration threshold.

The threshold of 30% has been used because it provides a quantitative value for comparison between communities, although it is difficult to determine what the best threshold in terms of ease of water or ice transportation is because of the grid cell resolution at 25 km and boat/snow machines are much smaller than this. The freeze-up and break-up day trends are similar if the sea ice concentration threshold that defines them is increased or decreased by 15% from the 30% threshold used in this analysis.

## 3 Results

### 3.1 Timing of freeze-up and break-up

The freeze-up day for Kotzebue Sound shows a much weaker trend than the freeze-up days for Barrow or Shishmaref (Figure 2). The linear trend of the date of freeze-up is a delay of 2.2 days per decade for Kotzebue and 6.0 days per decade for Shishmaref. In the early part of the record, there were years in which the maximum ice concentration near Barrow did not fall



below 30%. To derive as complete a record of break-up dates as possible, we estimated the date of break-up in these cases based on the dates at which ice concentration fell below either 45% or 60% using linear relationships determined from other years.

Considering break-up timing, Shishmaref again shows about a three times stronger trend than Kotzebue, with Shishmaref

seeing an earlier break-up by 3.4 days per decade, and Kotzebue at 1.1 days per decade (Figure 3). The timing of break-up and freeze-up trends in the Bering Strait (not shown) are 4.9 days per decade and 5.0 days per decade. The linear trend of the date of freeze-up was not calculated for Barrow, due to a relatively higher error from our interpolation of 30% freeze-up dates in earlier years when the ice concentration never reached below 30%.

Not only are there differences in the trends of freeze-up and break-up different between the selected communities, but the

variance is different as well (Table 1). Kotzebue shows 132% of the variance of freeze-up day for Shishmaref, and 108% the variance of break-up day. Fractions of the variance that are explained by the trend are 6% and 2% for the freeze-up and break-up day of Kotzebue, and much higher for Shishmaref at 44% and 20% for freeze-up and break-up day respectively. It is important to understand how much the variance plays a role in the trend because this can give indication of the strength of the trend. If the variability is not large relative to the observed trend in break-up and freeze-up day, then we have a better chance

for determining how much change we can expect to see at any particular time.

## 3.2 Duration of open water

Following the differences in trends of freeze-up and break-up timing, open water duration varies considerably depending on location (Figure 4). Barrow shows the largest change from the first 10 to the last 10 years of the dataset (1953-1963 to 2003-2013). The next greatest increase in open water duration is Wales, followed by Shishmaref and then Nome. Due to the smaller

trends in freeze-up and break-up timing, it is not surprising that Kotzebue shows the least change in the number of open water days. Some variability in freeze-up and break-up dates has to do with changes in oceanic heat input. Since Kotzebue Sound is somewhat more isolated from the larger Bering Strait throughflow and changes there, this could be a reason Kotzebue Sound is showing less of a change than outside of the sound. In addition, changes in river discharge can also impact freeze-up date, with a deeper freshwater lens over the Sound hastening freeze-up under the same atmospheric forcing.

# 4 Discussion: Ramifications of direct impacts

## 4.1 Enhanced coastal erosion

It is well documented that a lengthened period of open water leaves Arctic shorelines more vulnerable to erosion from autumn storms (Barnhart et al., 2014; Overeem et al., 2011). Moreover, the number of storms along the northern Alaskan coast that take place during the open water period may be increasing (Figure 5). Parkinson and Comiso (2013) presented a quantifiable

variation of coastline exposure due to inconsistencies in trends of the delayed freeze-up and earlier break-up of sea ice, but this study analyzed years starting with the satellite era in 1979. By using HSIA data, we were able to quantify changes in the





duration of the open water period along various areas of the coastline since 1953, before the satellite data became available. It should be noted that Parkinson and Comiso (2013) use a different definition of open water season in the sense that we do not count those days of open water before the last date of break-up and after the first day of freeze-up. Although there are other factors that influence coastal erosion rates (e.g. permafrost extent, surface geology, wind direction), changes in the local

timing of freeze-up may be an indicator of how vulnerable a particular coastline is to erosion. Kotzebue Sound shows less of a change in freeze-up and breakup trends than the other communities examined (Figures 2 and 3). A possible reason that remains to be explored is that Kotzebue Sound is somewhat "protected" from the larger-scale ocean currents that flow on the external part of the Sound. It has been suggested that there is an increasing mean northward warm water transport through the Bering Strait throughflow (Woodgate, 2012), and the relatively separated flow of the Sound compared to the external currents could

contribute further to changes in trends of freeze-up and break-up. Changes in oceanic heat transport also have implications for the effectiveness of the water to erode thawing permafrost coasts, with warmer water being able to erode notches more effectively (Wobus et al., 2011).

It is important to keep in mind that other factors affect communities' increasing vulnerability of storm-induced erosion. For example, differences between the angle of predominant wind direction and coastline orientation are important for direct

impacts of storms. Future projections for policy and adaptation measures must be aware of these and other differences in threats between communities in order to be maximally effective. Such differences can impact decisions on, for example, the safest and longer-term placement of water treatment areas in coastal communities.

## 4.2   Impacts on travel for subsistence hunting

Stable landfast ice is required for on-ice travel, and indigenous residents of Arctic coastal communities have noticed changes in

the quality of sea ice available for travel and hunting, as well as how the ice breaks up (Kirk, 2013). The timing of break-up is important to consider to determine the mode of transportation and impacts upon subsistence hunting. For example, the presence of ice near shore helps reduce waves for small boat travel. It is important to note that the trend in the earlier timing of break-up for Kotzebue, Shishmaref, and Point Barrow shown in Figure 3 are based on an approximately 25 km x 11 km (0.25 x 0.25 degrees) resolution sea ice concentration threshold, and may not capture the finer-scale resolution required for landfast ice.

The trend toward earlier break-up also means there are fewer days to hunt whales since the start date of the subsistence whaling season is determined by a combination of tradition and the migration date of whales. Figure 3 shows the decrease in the number of days left for whaling in Barrow, based on a nominal start date of April 15. The end of whaling season does not necessarily coincide with the break-up of the landfast ice or the retreat of ice from the coast. However, the traditional spring hunt in communities such as Barrow has relied on the ability to safely reach the edge of the landfast ice where seal skin boats

(umiaqs) can be launched to chase nearby whales (Druckenmiller et al., 2013; Gearheard, 2013). With earlier spring break-up of sea ice, the time window for this traditional method of whale hunting is becoming shorter and hunters must increasingly rely on aluminum boats with outboard engines. Further work extending these findings could include quantitative evidence to reveal whether or not a delay from hunting via boat to snow machine actually changes hunting success.




In some cases, wave heights hinder hunting success much more than prey abundance due to lack of access to the prey. Hansen et al. (2013) report that bowhead whale hunters consider prey abundance to be less of an issue than their own access to the prey due to the increased cumulative wave heights caused by increased open water fetch. Hansen et al. (2013) examined wind speeds recorded at Barrow in conjunction with threshold wind speeds reported by Wainwright hunters which were deemed

unsafe hunting conditions, and determined that a 11% shorter hunting season for Bowhead whales for spring and 12% for summer since 1971.

The impacts of an earlier sea ice break-up near the coast and offshore are difficult to predict due to the possibility of effective adaptation. For example, if whalers can successfully hunt from the beach instead of the landfast ice edge, as they do during the fall whaling season, then perhaps their hunting success would not decrease so drastically. In terms of handling larger waves

that are due to increased fetch and the increasing amount of time open water is exposed to storm activity, perhaps hunters can obtain access to more stable, suitable, and bigger boats. The impacts of climate change and changes in the timing of freeze-up and break-up might be less severe if communities are able to adapt their hunting practices effectively.

## 4.3   Prey availability

The subsistence harvests of all Arctic coastal communities include ice-associated marine mammals such as ringed and bearded

seals, walrus, belugas, and bowhead whales (Moore and Huntington, 2008). The availability of each of these species, in terms of both the population size and proximity to hunting grounds, is likely to be impacted by changes in ice conditions related to the timing of freeze-up and break-up. In this sense, future prey availability could be indicated by the freeze-up and break-up trends seen in the HSIA data (Figures 2 and 3). Species reported to be particularly sensitive to ice conditions are walrus and bowhead whales. For a later freeze-up, walrus have had to shift where they give birth so that they have can have access to the

sea ice (Burns, 1970). Work done by Kapsch et al. (2010) identified optimal conditions for maximum walrus hunting success in St. Lawrence Island, Alaska, of 0 to 30% ice concentration, and specific windows of wind speeds, temperature, and visibility. This suggests that a delayed freeze-up will shift the optimal walrus hunting conditions to later in the year, though Kapsch et al. (2010) found that hunting success was more sensitive to ice conditions in spring than fall.

Other shifts in hunting are occurring with changes in the Bowhead whale migration, because Bowheads are arriving earlier to

the Beaufort in August (Clarke et al., 2014). Their migration is influenced by polynas and leads (Lowry, 2000). An indication of changes in the time window of migration for whales through the Bering Strait could be the increase in number of open water days there (Figure 6).

Ringed seals have shown a decline in health which have been driven by changes in persistence and extent of seasonal sea ice (Harwood et al., 2015). Ringed seals are ice-associated year-round, and breeding occurs on stable ice with good snow cover. If

a delayed freeze-up (Figure 2) means there is less time for the snow cover to develop, pups are more exposed to predators due to inability to construct an adequate lair (Kovacs et al., 2011). Bearded seals are likely to be negatively impacted by an earlier break-up (Figure 3), because they require stable seasonal ice late in the spring for raising pups and moulting (Kovacs et al., 2011).



If the migration changes of animals that have been hunted for generations contribute to problems in number of catch, adaptation solutions must be found. It has been noted by communities in Nunavut, Canada that the delayed freeze-up results in thinner ice, delayed seal hunting, altered travel routes, and more difficulty locating the seals (Laidler et al., 2010). Changes in timing of freeze-up and break-up can also have consequences in willingness to take risk due to the shorter season (Figure 3) for hunting and travel over the ice (Ford et al., 2006).

Communities with diverse subsistence harvests might be less affected than those that depend on one or two primary sources. The St. Lawrence Island communities of Savoonga and Gambel might be unfortunate examples of the latter. They are highly dependent on walrus harvest, which has been significantly impacted by sea ice changes in recent years, with static regulations for hunting in conjunction with seasonal changes can influence hunters to travel further in more dangerous conditions (Fidel et al., 2014). Not only subsistence hunting, but travel between communities is impacted by the timing of freeze-up, and this can affect trade and exchange of subsistence catch and goods. This could limit availability of subsistence animals which are not harvested from hunters of one particular community, but which must be traded between another community.

## 4.4 The importance of a changing transition season

A transition period can be defined as the portion of the seasonal sea ice cycle between the time of open water and frozen ocean. Here, we define the freeze-up transition period as the time it takes for open water (0% sea ice concentration) to freeze to a 30% sea ice concentration. Similarly, the breakup transition period is defined here as the time it takes for 30% sea ice concentration to decrease and become open water. For subsistence communities, the start of the freeze-up transition period means a closing window for safe transportation via small boats. After the freeze-up transition, once landfast sea ice has formed it becomes safer to travel to hunting areas or other communities by snow machine, dog-sled, or regular street vehicles. The spring transition period has been showing an earlier break-up (Figure 3), and, in some areas of the Arctic, a longer transitional stage has been reported. These reports show uneven ice deterioration which alter travel routes, causing people to be "stuck" in town longer with hampered access to hunting grounds (Laidler et al., 2009). There has been much anecdotal evidence from community members across the Arctic that suggest the transition periods are becoming more prolonged. Ross Schaeffer from Kotzebue observes "we have a longer fall and a longer spring so it's warming on both ends and the winter is getting shorter ... used to be everything would melt in one week" Schaeffer (2013). The changing shoulder seasons interfere with the native subsistence way of life by altering travel routes, leading to less predictable sea ice conditions which could impact safety of the hunters. Changes in the ice conditions have been documented for several communities in Nunavut, including Igloolik, Pangnirtung, and Cape Dorset (Laidler et al., 2008; Laidler and Ikummaq, 2008; Laidler and Elee, 2007). For example, in Igloolik, elders have noticed both an extended freeze-up period, and an extended break-up transition length:

"What [I] have noticed...in the last five to eight years, [is that] when it should be freezing up...it becomes overcast, snow starts falling for a long period of time...So that affects freeze-up...In that whenever it's overcast the temperature rises a bit, freeze-up doesn't occur as quickly, or it doesn't even occur at all at some times when you have clouds and the wind working together" (Aqiaruq, 2004b).



Comments like this highlight the complexity of the transitions between open water and ice-covered seasons. It therefore might be problematic to simply use a sea ice concentration threshold to define the shoulders of the break-up and freeze-up transition seasons. We examined variability in the duration of the 0-30% period, but found no significant trends.

## 5    Discussion: Ramifications for globally-induced local impacts

### 5.1    Regime shift in sea ice navigability: delivering goods and increased Western influence

As a result of events involving sea ice hazard, for example as in the Beaufort Sea when there was an unusually high amount of ice along the coast in 1975 which caused problems for goods delivery (Heimer et al., 1978), there was an increased interest in predicting future sea ice anomalies. Barnett (1976) summarizes what was immediately done in terms of developing a long-range forecasting method motivated by the 1975 severe ice conditions. It was found that there were relationships between April meteorological conditions in Siberia and the Alaskan Arctic and the ice conditions during summer in north Alaskan shipping lanes. Based on these relationships, a long-range (e.g. from April to summer) method for forecasting sea ice was developed with a severity of ice conditions index (BSI) from 1953-1975 (Barnett, 1976) and 1979-2000 (Drobot, 2003).

We have extended the BSI values to include more recent years, filling in the gap from 2001 through 2013. Our BSI values are calculated from the HSIA dataset, and cover the years from 1953 through 2013. The calculation for this index includes the distance from Barrow to the sea ice edge, and characteristics of ease of shipping from Prudhoe Bay to the Bering Sea (Drobot, 2003). The severity indices of Barnett (1976) and Drobot (2003) correspond well with the number of open water days calculated from HSIA data (Figure 7). Some years Barnett (1976) calculated having a low BSI (unfavorable ice conditions) with a minimum number of open water days, while HSIA data shows no open water days, and so a corresponding HSIA BSI index of 0. This can be seen in the vertical linear feature near the HSIA BSI values of 0 in Figure 8. For example, in 1955, Barnett (1976) calculated a BSI index of 50 (still indicating a very hazardous ice year), while the BSI we calculated here from HSIA data is zero (Figure 7). In the extreme high ice hazard year of 1975, both (Barnett, 1976) and HSIA BSI values are found to be 0. An explanation for discrepancies in sea ice concentrations and therefore BSI values is that other datasets were incorporated into HSIA concentration values besides the ice charts used by Barnett (1976). Further, the spatial resolution for fetch Barnett (1976) used (5 nautical miles) is much higher than the spatial resolution the HSIA is able to capture ( 25 km or 13.5 nautical miles).

Recent rapid changes in the length of open water season at Barrow (Figure 7) have led to a "regime shift" for maritime traffic destined for Barrow or locations further east (Smith and Stephenson, 2013), making a repeat of the events of 1975 now seem virtually impossible. Although there is large interannual variability in the number of open water days, there is a marked increase in open water duration. After 1992, all years show at least some ice-free days (ice-free defined here as less than 30% sea ice concentration). Not only at Point Barrow, but longer navigation seasons along other Arctic coasts is very likely, which is leading to an increased use of coastal shipping routes and development of offshore continental shelves (Instanes et al., 2005). Associated political, economic, and social consequences for all residents of Arctic coastal areas could be significant





and possibly outweigh direct physical impacts from global warming. For example, increased development of coastal shipping routes could lead to an increase in the availability of non-subsistence food and goods, and also social changes such as an increased number of associated mixed-wage opportunities. These combined indirect effects may be more significant than the direct consequences of an increased open water season on subsistence hunting activities.

The Bering Strait, which is a key area for Arctic shipping, also shows a significant upward trend in the number of open water days per year (Figure 6). The five-year running mean of open water days in the Bering Strait shows an approximately 70 day increase in the number of open water days from when the dataset began in 1953. The lengthened open water season has implications for vessel traffic because when the time window of open water is narrow, quick decision-making and accurate prediction of sea ice movement becomes vital for navigating through sea ice hazards (Whiteman, 2011). The break-up and

freeze-up timing of the Bering Strait is important for vessel navigability, and when they can leave port for travel through the Strait.

The United States Coast Guard (2013) reports a 118% increase in maritime transit through the Bering Strait from 2008 to 2012. Recently, the US Coast Guard has released a recommended shipping route through the Bering Strait (United States Coast Guard, 2017), that will can be used by vessels transporting goods, as well as a new tourist cruises made possible by

the lengthened open water season. This defined shipping route away from the coastline could help alleviate hazards from potential oil spills, and make it easier to regulate speed limits and traffic numbers so as to minimize interference with the many marine subsistence mammals which use the Strait for their annual migration (Braham, 1980; Sakakibara, 2011). It is difficult to determine what impacts increased vessel traffic will have on the food, water, and energy security of coastal communities because the changes are happening very fast. Food security could be impacted in the sense that increased goods delivery

might influence certain coastal communities or individual members of those communities (for example, young people) to more heavily rely on outsourced food products, instead of relying on subsistence.

**5.2   The importance of knowledge sharing from different societal levels**

Cumulative impacts from a lengthening open water season are being seen on a variety of societal scales, cascading from the high levels of government and well-resourced private companies to local interactions between members of a rural subsistence

community. The decision to drill for oil offshore can provide coastal communities with jobs, but can also change access to hunting grounds. It is this inevitable connectedness between high levels of government and local community interaction that necessitates efficient sharing of knowledge on all societal scales. Due to the predominant direction of the cascade (e.g. from high-level large government bodies down to impacts on indigenous Alaska communities) it is difficult for knowledge to transfer from the community-level upwards. Knowledge sharing on different societal levels has been referred to as "institutional

interplay" and has been deemed essential to the collective ability of people to manage environmental change (Young, 2002). Incorporation of knowledge from the physical science basis to all levels of the societal system is difficult because it requires crossing boundaries of discipline-specific scale perspectives (Abdelati et al., 2010). The HSIA is an example of a dataset useful for all parties to examine large-scale and decadal changes in sea ice, because it is accessible (especially since it has the quick-glance graphical user interface option). It is therefore very important to support studies and efforts that are able to identify key





temporal and spatial scales relevant and attractive to multiple research disciplines. This remains challenging because thresholds are sensitive to scale (Abdelati et al., 2010) and can vary in importance depending on which perspective is taken in the coupled physical-ecological-human earth system.

The fact that traditional Arctic communities have existed for centuries means a successful understanding of the threats present in the harsh Arctic climate, sustainable resource use, and an understanding of how natural resource availability varies over time. Knowledge from these communities is therefore indispensable for large global actors such as government bodies and multinational companies to understand how their own actions will impact the local ways of life. An example of progress being made on this front is when visitors to communities rely on local knowledge for risk assessment, such as when researchers visit a community and are given advice about where and when is safe to go out on the ice.

## 6 Conclusions

Using the HSIA quarter-monthly dataset we have evaluated changes in the local sea ice regimes near Alaska coastal communities and at other key points of interest for commercial maritime activities over the period 1953-2013, such as extending the Barnett Severity Index (BSI) calculated timeseries. The BSI is useful because it provides a metric for ease of shipping derived directly from sea ice conditions. Examining how this has dramatically changed over recent years provides a context for evaluating future change in Arctic communities.

Our analysis reveals a regime-shift in the number of open water days at Barrow, as well as differences in the timing of freeze-up and break-up among Alaskan coastal communities. Freeze-up was shown to have more of a delay in Shishmaref than Kotzebue with freeze-up occurring later by 6.0 days per decade and 2.2 days per decade, respectively (Figure 2). Shishmaref also showed about a 3 times stronger trend in timing of earlier break-up timing, given an earlier break-up in Kotzebue of 1.1 days per decade and 3.4 days per decade in Shishmaref (Figure 3).

Earlier break-up in Barrow has resulted in a much shorter season for the traditional spring whale hunt since 1953, roughly halving it (Figure 3). With earlier coastal break-up and more open water, whalers are having to use less traditional and potentially more costly means of harvesting bowhead whales in Barrow. Impacts on marine mammals important for subsistence include changes in bowhead whale migration, and impacts on breeding locations for walrus, ringed and bearded seals.

We have identified two types of impacts on Alaskan coastal communities likely to result from these changes in sea ice: indirect and direct effects. Further quantitative comparison of direct versus indirect impacts is left for future work, although we expect the balance of impacts will be locally specific and depend upon the severity of sea ice change, coastal morphology, local hunting practices and connectedness to the global economy. The extended open water season allows for indirect impacts stemming from globally-induced economic decisions, such as more populated and new shipping routes as well as increased tourism. Direct impacts from reduced sea ice presence come from changes in timing of freeze-up and break-up, which have been documented to impact safety of travel and access to hunting grounds.

The sea ice regime variables of the timing of freeze-up and break-up have the greatest impact on local activities. Knowledge about the differences between direct local impacts that are induced by a changing physical climate and indirect impacts associ-



ated with the global economy is important in terms of policy decision-making for adaptation measures. One concrete example clearly illustrating the fact that policy-makers must be aware of changes in freeze-up and break-up timing is that while hunting regulations (e.g. legal hunting dates) remain static, the changes in timing of freeze-up and break-up in conjunction with climate change-induced migration shifts leave reduced time available for subsistence hunting, encouraging hunters to go out on thin

5   ice or in unsafe conditions, evaluating risk differently than they did before.

It is also important to keep in mind that changes in the lengthened open water period are not uniform throughout Arctic communities, and we can therefore expect varying consequences. This work has highlighted immediate research priorities for the future, such as the need to develop quantitative metrics for freeze-up duration that are practical to the community, and quantitative evidence of changes in hunting success and erosion rates that are direct results from changes in freeze-up and

10  break-up timing. Certain methods should be developed to help assess the relative impacts of direct versus indirect impacts. One such method could be to include a way to quantify hunting effort versus economic cost. Another could be to assess social and economic contribution to local economy from oil development and tourism revenue against the cost of a potential oil spill. With recent rapid and future sea ice change, being aware of and quantifying direct and indirect impacts is an important step for future adaptation of coastal communities.

15  *Data availability.* The dataset used in this work is the Historical Sea Ice Atlas, available for download at $http://data.snap.uaf.edu/data/Base/Other/Historical_Sea_Ice_Atlas/$

*Competing interests.*  The authors declare that they have no conflict of interest.

*Acknowledgements.*  This work was supported by National Science Foundation award 1263853 for the Sustainable Futures North Project.





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




Selected Grid cells used for analyzing sea ice concentration

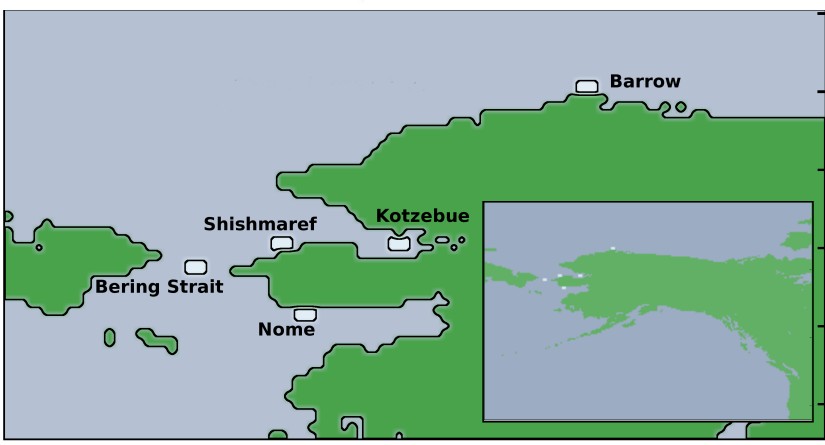

**Figure 1.** Selected grid cells from Historical Sea Ice Atlas used to analyze sea ice concentration, freeze-up, and break-up days in several communities and one location offshore (Bering Strait).



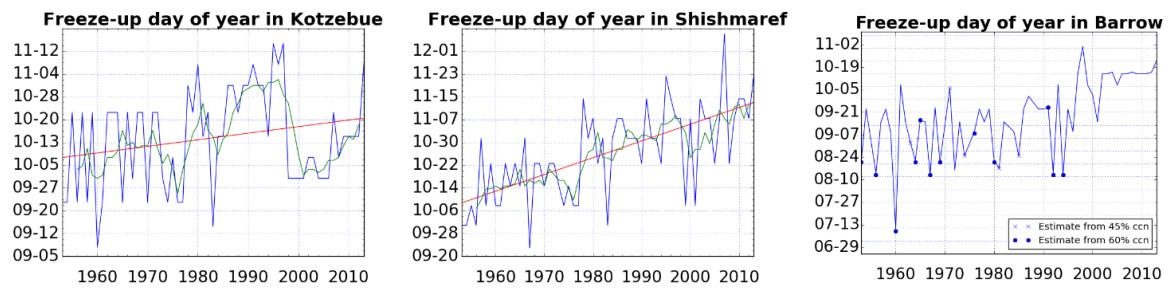

**Figure 2.** Trends in freeze-up day of year for Kotzebue and Shishmaref. Blue line is yearly freeze-up day, and green line indicates the 5-year running mean. For Barrow, some years did not reach below a 30% sea ice concentration, so freeze-up days for these years were calculated using linear regression.



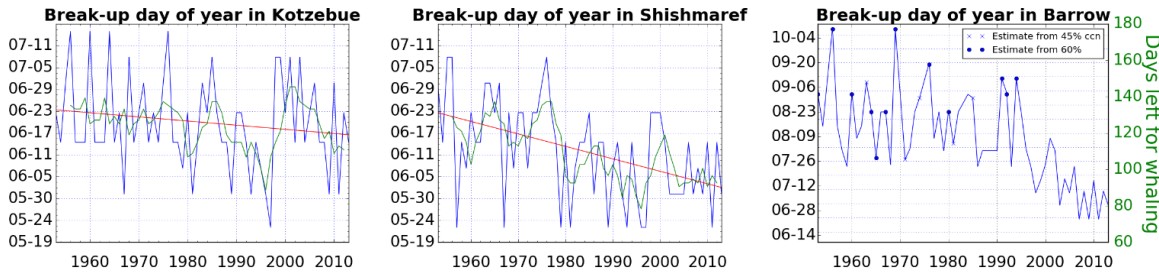

**Figure 3.** Trends in break-up day of year for Kotzebue, Shishmaref, and Barrow. Blue line is yearly freeze-up day, and green line indicates the 5-year running mean. For Barrow, some years did not reach below a 30% sea ice concentration, so break-up days for these years were calculated using linear regression.



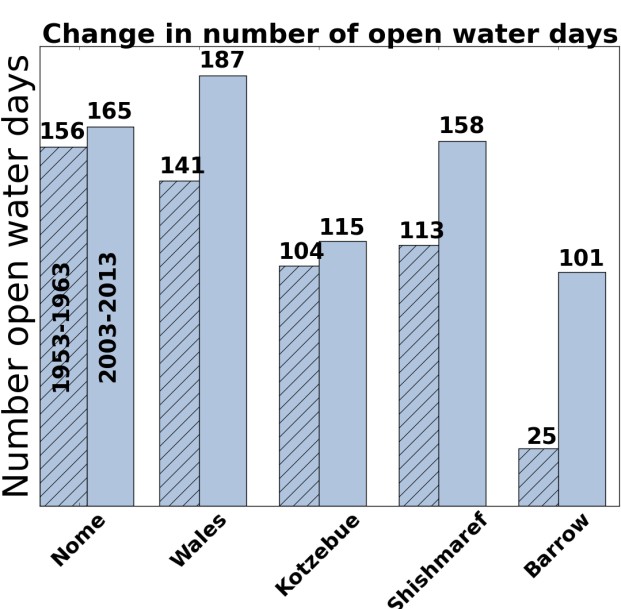

**Figure 4.** The increase in open water period varies between communities. Hatched lines show the mean number of open water days (summer period below 30 percent sea ice concentration) for the first 10 years of the Atlas dataset (1953-1963), and the unhatched bars show the last 10 years of the dataset (2003-2013). We can expect the consequences to be different between communities.





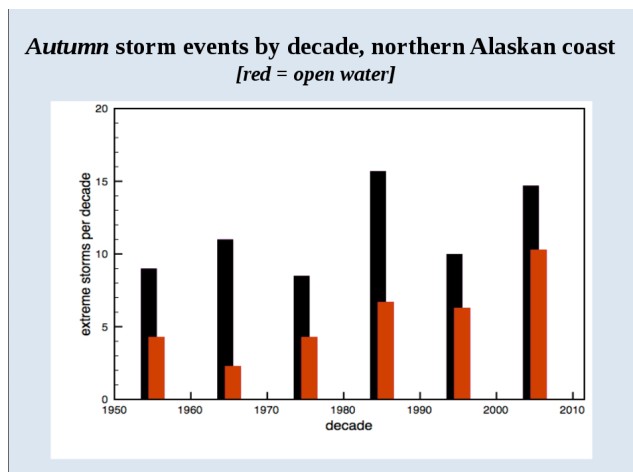

**Figure 5.** The number of extreme fall storm events during the open water period may be increasing along the northern Alaskan coast. Extreme storms defined here are those events in the top 1 percent of pressure gradient difference. Data is from NCEP Reanalysis. Figure provided by William Chapman, University of Illinois.





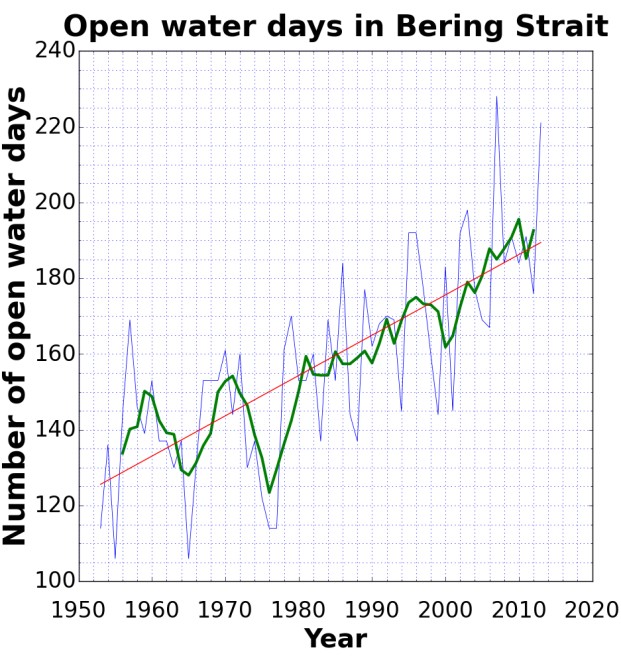

**Figure 6.** Number of open water days in the Bering Strait shown along with the trend line (red) and as a 5-year running mean (bold green line).





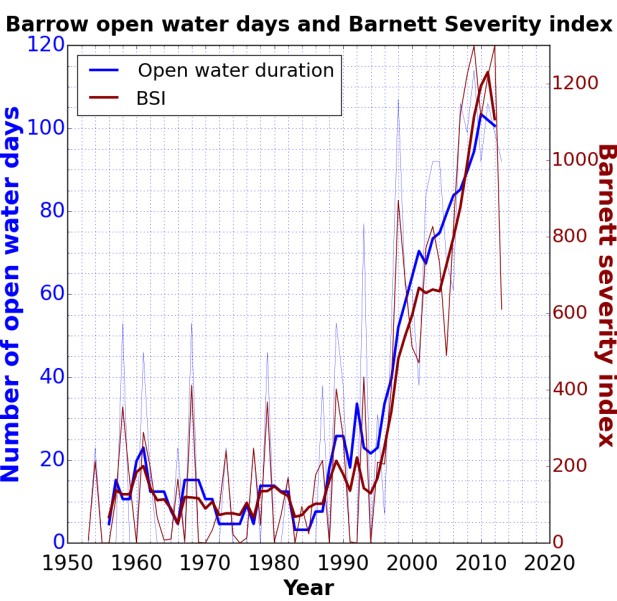

**Figure 7.** Number of open water days at Barrow, AK has increased substantially since the dataset began in 1953. There has been a 'regime-shift' such that after 1992, Barrow has seen at least some days of open water. Bold red and blue lines show the 5 year running mean of the BSI and the number of open water days as defined by a 30 percent sea ice concentration threshold, respectively. Barnett Severity Index (BSI) agrees well with increasing open water trend (lower values indicate higher severity ice years)



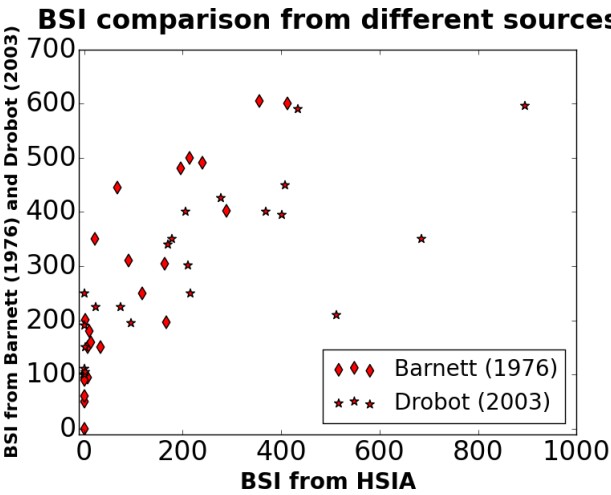

**Figure 8.** Comparison of Barnett Severity Index (BSI) calculated from Barnett (1976), Drobot (2003), and the Historical Sea Ice Atlas (HSIA)





**Table 1.** Variance of freeze-up and break-up trends

|  | Trend [days/year] | % variance explained by trend |
| --- | --- | --- |
| Freeze-up Kotzebue | 0.23 | 6 |
| Freeze-up Shishmaref | 0.60 | 44 |
| Break-up Kotzebue | -0.10 | 2 |
| Break-up Shishmaref | -0.34 | 20 |