# Peer review of "Impacts of a lengthening open water season on Alaskan coastal communities: deriving locally-relevant indices from large-scale datasets and community observations"

_The Cryosphere, 2017_

## Referee Comment (RC1) · Anonymous Referee #1 · 31 Oct 2017

Review of: Impacts of a lengthening open water season on Alaskan coastal communities by Rebecca J. Rolph, Andrew R. Mahoney, John Walsh, and Philip A. Loring

Summary:

This paper aims to use the Historical Sea Ice Atlas (HSIA) to assess potential direct and indirect impacts from sea ice change for 3 (or 4) Alaska communities. Unfortunately, the analysis described does not sufficiently support the paper's conclusions, due to a lack of focus and a clear connection to the primary analysis. The paper simply tries to cover too much ground. Further development, extension, and quality-control of the HSIA analysis is recommended, with additional explanation of the physical context of the sea ice environment at study locations, and more explicit linkages to community interaction and feedback regarding key impacts.

[Figure]

Review metrics:

1. Does the paper address relevant scientific questions within the scope of TC?

Yes, but it is encumbered by side issues.

2. Does the paper present novel concepts, ideas, tools, or data?

Partly, but not fully realized.

3. Are substantial conclusions reached?

No.

4. Are the scientific methods and assumptions valid and clearly outlined?

No – though if better explained and more thoroughly executed they may be.

5. Are the results sufficient to support the interpretations and conclusions?

No – the discussion section relies too little on the results of the analysis.

6. Is the description of experiments and calculations sufficiently complete and precise to allow their reproduction by fellow scientists (traceability of results)?

Only in part. The method of recalculating the Barnett Severity Index (BSI) was not given.

7. Do the authors give proper credit to related work and clearly indicate their own new/original contribution?

Mostly.

8. Does the title clearly reflect the contents of the paper?

The title is nominally accurate but the paper does not succeed due to overreach.

9. Does the abstract provide a concise and complete summary?
Improvement is needed.

10. Is the overall presentation well-structured and clear?

No.

11. Is the language fluent and precise?

Improvement is needed, especially toward the use of plain language instead of jargon.

12. Are mathematical formulae, symbols, abbreviations, and units correctly defined and used?

Yes.

13. Should any parts of the paper (text, formulae, figures, tables) be clarified, reduced, combined, or eliminated?

The organization of the paper is poor, it attempts to cover too much ground on the basis of too little new analysis, and it would be improved by substantially editing the discussion section down to those questions clearly related to the HSIA analysis. The quality of the figures is low.

14. Are the number and quality of references appropriate?

References for recent sea ice research papers are lacking; references to similar studies on indigenous knowledge of sea ice for the St. Lawrence Isl. communities, for example, are incomplete.

15. Is the amount and quality of supplementary material appropriate?

N/A

Abstract:

The abstract could be more concise with the removal of some unnecessary phrases like 'It is often remarked' (Pg1 L1) which only weakens the statement that 'Alaska coastal

communities are on the frontlines. . .' Also the phrase 'navigational regime shift' (Pg1 L11) is not the best choice given the freighted meaning of the term 'regime shift' in the physical sense, and further obscured by reference to 'navigational'. One could simply state that the impact of ice conditions on vessel operations near Utqiavik (Barrow) have eased since. . .[because]. The term 'regime shift' here and elsewhere is an example of jargon that doesn't add materially to the paper and should be avoided as it is not essential to the work at hand.

Introduction:

It would be helpful to mention the role of the Barnett Severity Index (BSI) since it is highlighted in the abstract. It is also reasonable to provide the specific date range of the study, and a general sense of the application of "community feedback and inter-action" with the research team, or, if indirect, the sources of indigenous knowledge incorporated in the study (i.e. collected interviews of elders, joint review of archived interviews, previous published work, etc.). It may also be appropriate to mention work by I. Krupnik, for example, that is of a similar nature. The introduction would also be a good place to mention the specific communities discussed in the paper (note that the community of Wales is introduced in results section 3.2 for the first time).

For this paper an organization statement in the last paragraph of the introduction would be useful.

Data and Methods:

The structure and completeness of this section could be improved by separating and briefly addressing each element identified previously, for example:

a) HISA and its underlying data and related issues/benefits for the specific period of the study (e.g. blend of surface and satellite obs, use of microwave data near the coast, impact of quarter-monthly resolution);

b) sources of indigenous knowledge used in the study and how they were used;

c) selection of study areas and reference HISA grid cell locations in light of (b) or otherwise;

d) selection of metrics, as in 30% ice concentration threshold (Serreze et al. 2016 doi:10.1002/2016JC011977 provide a good rationale);

e) BSI, which is absent in this section and the introduction but mentioned in the abstract – this is especially important as the BSI index uses sea-ice inputs only for the area north of Barrow and along the coast to Prudhoe Bay, hence its extension to distant study areas should be justified, and particularly given Barnett and Drobot's cautions regarding use of historical data prior to ∼1978 and the impacts of potential ice loss on the utility of the BSI; losses that have now taken place.

f) some brief general description of how direct and indirect impacts will be evaluated in light of results (e.g. erosion, travel for hunting, prey availability, transition season, as well as indirect ones) would be useful here.

There are elements more appropriate for the introduction and/or discussion section included here that should be removed (e.g. line 13 and following).

Results:

The issue with Barrow in the first paragraph requiring an alternate calculation for break-up date should be explained in the methods section, and especially justifying the linear interpolation. It appears that in some years the ice never moved back sufficiently from the coast, and thus the interpolation leads to the inference of a condition that may have never physically existed. Inspection of the pertinent variables given by Barnett (1976) show 11 years (of 23 examples) 1953-1975 where the sea ice did not retreat at all (as measured in nautical miles), or not far enough to be detected in the reference grid box (esp. 1975).

Interpretation of these results depends on the physical context that helps explain why differences exist between the various study locations. For example, ice cover near

Wales is influenced by the dynamics of the Bering Strait inflow, and local/advected solar heating in this and especially the Alaska Coastal Current (ACC). The ACC is not well observed by the Bering Strait mooring array (Woodgate 2012 and related papers) and should probably be considered by reference to additional sources. The Kotzebue area is dominated by fast ice, influenced by local solar heating, river discharge plumes and related freshwater stratification, and factors related to the ACC at the outer edges, and is probably less sensitive to dynamics (i.e. ocean currents, wind-driven flaw leads, etc.) until later in the break-up process, compared to other locations. The amount of open water in Barrow, until recent decades, was defined primarily by the width of the lead between the coast/fast ice edge and the heavy polar ice pack (e.g. Hunt & Naske 1977), which is in turn influenced by (among other factors) large-scale atmospheric patterns such as identified by Barnett (1976) and others since. An important factor at Barrow then may be the major loss of multiyear ice in the recent period, and accompanying changes in the mobility and strength of the ice pack. This is perhaps the most important distinction between Barrow and other locations where first-year ice has been the norm over the entire study period.

The organization of the results section could be improved, perhaps by treating each phenomena as a sub-section, and by adding some analysis of the physical context required to understand why the observed differences may exist. This may well throw light on how communities' experiences and responses vary in the face of change to be discussed in following sections. Most sections may benefit from an opening remark explaining what follows and its organization (this is often done in the last paragraph of the introduction as well).

Discussion:

The connections between the results of the analysis of HSIA data and the generally vague and unfocused discussion sections are unclear. Finding that there have been various changes in the timing of break-up, freeze-up, and open water duration is not novel in itself, though the impact on communities may be, if better explained by the

specifics of the HSIA analysis here and its linkage to indigenous knowledge.

At points in the discussion the authors undermine their purpose with statements like: "Comments like this highlight the complexity of the transitions between open water and ice-covered seasons. It therefore might be problematic to simply use a sea ice concentration threshold to define the shoulders of the break-up and freeze-up transition seasons. We examined variability in the duration of the 0-30% period, but found no significant trends" (Pg 8 L1). This is rather discouraging to the reader.

The BSI is introduced in discussion section 5, where the authors describe extending the index from 2000 to 2013, but do not explain specifically how or why (which should have been done in data and methods if this was an important component of the paper). Then they report that the original index and the extended index (HSIA BSI) don't agree very well and this may be due to basic incompatibility of the input data. The BSI is not mentioned again until one sentence in the conclusion, which begs the question of why it was in the paper at all. It would be interesting to know if the BSI is still used for long-range sea ice forecasting (its original purpose) or for other reasons, but its inclusion in this paper is not warranted.

Conclusions:

Relationships between the specific HSIA-based sea ice analysis reported here and impacts of a lengthening open water season on Alaskan coastal communities are not clearly shown, especially with respect to community interaction and feedback.

---

## Referee Comment (RC2) · G. Ljubicic (Referee) · 30 Nov 2017

- Does the paper address relevant scientific questions within the scope of TC? Yes.

- Does the paper present novel concepts, ideas, tools, or data? Yes.

- Are substantial conclusions reached? They need to be better supported.

- Are the scientific methods and assumptions valid and clearly outlined? See comments in attached .pdf.

- Are the results sufficient to support the interpretations and conclusions? See comments in attached .pdf.

- Is the description of experiments and calculations sufficiently complete and precise to

allow their reproduction by fellow scientists (traceability of results)? In relation to HSIA analysis, yes. In others, need to be better supported.

- Do the authors give proper credit to related work and clearly indicate their own new/original contribution? Yes, but could be better referenced.

- Does the title clearly reflect the contents of the paper? Yes.

- Does the abstract provide a concise and complete summary? Yes.

- Is the overall presentation well structured and clear? Yes, but needs a bit more context in places.

- Is the language fluent and precise? Yes.

Are mathematical formulae, symbols, abbreviations, and units correctly defined and used? N/A

- Should any parts of the paper (text, formulae, figures, tables) be clarified, reduced, combined, or eliminated? See comments in attached .pdf.

Are the number and quality of references appropriate? See comments in attached .pdf.

Is the amount and quality of supplementary material appropriate? Yes.

Please also note the supplement to this comment:
https://www.the-cryosphere-discuss.net/tc-2017-211/tc-2017-211-RC2-supplement.pdf

**Supplement:**

**Impacts of a lengthening open water season on Alaskan coastal communities**

Review for "The Cryosphere" journal, Discussion Paper

November 30, 2017

The premise of this paper is to highlight the impacts of a lengthening open water season on Alaska coastal communities. Overall I find it an interesting and highly relevant paper. There is much current interest in defining impacts of environmental change according to indicators of relevance in a community context, and thus ultimately to help support decision-making at different scales. The extended record of the Historical Sea Ice Atlas (HSIA) is a valuable dataset for analyzing trends over time, although there are multiple challenges in doing so at appropriate scales and within particular community contexts. This paper tries to cover a lot of ground in a short paper. There are a number of good points made, although several areas where I would like more clarification, appropriate reference support, and nuanced discussion. It is also a paper well suited for discussion, and I look forward to reading feedback from other reviewers and discussants. Below are my contributions to this iterative review process, organized according to the key areas where I feel revisions would be needed prior to acceptance for full publication.

1) **Community uses of sea ice** – Given the premise of the paper to investigate the direct and indirect impacts of a lengthening open water season on four Alaskan coastal communities (Barrow, Kotzebue, Shishmaref, and Nome), I think it would be important to have more characterization of geographic (e.g. physical conditions, typical sea ice extent and cycles) and cultural (e.g. uses of sea ice particular kinds of sea ice for particular hunting or harvesting practices, links to seasonal traditions or community events) context up front. It could be added to the introduction, or be in a new "community context" section, but without this it makes it hard to interpret some of the arguments being made later in the Discussion. The references to communities are highly generalized, without much sense of how their uses or priorities for sea ice may be shared or different, and this would help to strengthen arguments as well as deepen the relevance of the analysis to the communities in question. Related to this, there is little explanation for the selection of the four communities beyond the diversity of sea ice regimes and subsistence activities (but these are not really introduced). In addition, Nome does not appear in most trend analyses, and Wales appears inconsistently in the text. More explanation and consistency in the communities of interest would be important. Furthermore, many of the references I am used to seeing that describe community use, conditions, and importance of sea ice from local perspectives in Alaska could be better incorporated throughout this paper to support both the local context as well as the analysis of direct and indirect impacts (e.g. work by Huntington, Eicken, Krupnik, Norton, George, Druckenmiller, among others).

2) **Selection of 30% ice concentration threshold** – The choice of selecting 30% threshold for freeze-up and break-up needs more discussion and justification, as well as greater consideration of associated limitations. I find this to be quite low, if considering travel on landfast ice. It would also be very helpful to more clearly relate this to community use of sea ice. What would local perceptions of freeze-up approximate to in terms of ice concentration? You cited some of my previous work (Laidler et al., 2009) in which we used 9/10 (90%) ice concentration for freeze-up in terms of being navigable on snowmobile or foot (vs. 5/10 which is the common definition for freeze-up in relation ship navigation). At 30%

concentration I would think there is still a lot of broken moving ice. Perhaps the overall trends would not change much, but this threshold selection is critical in terms of the arguments being made, and how this would translate to impacts on communities. This also has important implications for how transitional stages are considered, which are not really captured with one threshold (e.g. 30% used as break-up and as ice-free definition within the paper). I think this threshold selection and representation of transitional seasons is deserving of more careful consideration and/or articulation.

3) **Figure 1** – This figure does not give a good sense of scale of ice area covered around each community, or resolution of grid cells. Could a larger and more detailed figure be created to better represent this?

4) **Interview citations** – Two interviews are cited in the text, and referenced as being interviewed in Kotzebue in 2013. There is no other context about these interviews in terms of how they were related to this research or other community-based projects, or any details in the Methods section about how interviews were undertaken and with what focus and which participants. I would like to see more of these local and Indigenous perspectives included in the paper, but they also need to be clearly explained and included in methods. Furthermore, interview quotes included from other papers need to be fully cited to the paper they were published in (as well as the individual), so they can be appropriately credited and contextualized.

5) **BSI interpretations** – This analysis does not seem well connected to the rest of the paper, and the calculations and methods involved are presented in the Discussion rather than Methods. Perhaps getting into this analysis in sufficient depth is beyond the scope of the paper? It would be good to really clarify what the primary goal and emphasis of the paper is. If it is indeed on community impacts (and related to community priorities and concerns), then expanding in areas noted above may be preferred to this particular aspect of analysis.

6) **Societal levels and accessibility arguments** – I think what you are trying to refer to here is not societal levels (or scales), but decision-making or jurisdictional scales. This needs to be clarified throughout. The arguments here are also covered in such a generalized fashion, that it is difficult to connect to the sea ice and community-specific trend analysis. What would this mean in different community contexts? And when you talk about the accessibility of HSIA, to whom are you referring? How accessible and useable (and/or currently used) is the HSIA in Alaskan coastal communities?

7) **Typos and References** – There are a number of minor typographic errors throughout, as well as a number of incomplete references, that need to be attended to. I can provide more details on these if requested.

In the process of trying to compile my feedback, a number of other questions have arisen for me. But I will leave it here to see what the other reviewers and discussants say, and how the authors choose to respond. I am then happy to continue being part of the iterative review and discussion process.

Best wishes,
Gita Ljubicic
Department of Geography and Enviornmental Studies
Carleton University, Ottawa, Canada

---

## Short Comment (SC1) · 30 Nov 2017

HJ van Meerveld

ilja.vanmeerveld@geo.uzh.ch

Dear authors (and editor),

In the "Current topics in Earth System Science" (ESS 401) course at the University of Zurich, we ask students to choose a manuscript that was recently submitted to one of the Copernicus journals and to write a review for this manuscript. Two students (Florian R. Lustenberger and Bastian Buman) chose your manuscript and wrote the attached review. We hope that you find it interesting and useful.

Best regards,

Ilja van Meerveld

[Figure]

Please also note the supplement to this comment:
https://www.the-cryosphere-discuss.net/tc-2017-211/tc-2017-211-SC1-supplement.pdf
* * *
[Figure]

**Supplement:**

**Summary of the manuscript**

The authors have used the Historical Sea Ice Atlas (HSIA) to calculate a date for the break-up and the freeze-up of the sea ice for four coastal Alaska communities (Barrow, Kotzebue, Shishmaref, Nome) as well as for an area in the Bering Strait. The dates were calculated from 1953 to 2013 based on a threshold of 30% ice-cover. Based on this data a linear trend was derived to find a (possibly climate change associated) change in the timing of both freeze-up and break-up.

Following this analysis, the paper reviews numerous potential interactions (direct as well as indirect) between the change in sea ice and the impacts on indigenous peoples.

**Main Assessments**

The study discusses a current topic related to climate change, namely the duration of sea ice cover.

However, it remains unclear how these communities were selected and why the data for not all communities that were selected (p. 2, l. 30) are presented (figure 2 and 3) and discussed (results, discussion). Additionally, the methods are lacking in detail and statistical details are not addressed. A significant flaw is the lack of an evaluation of the trend line. As the trend line is the main result of this study, it requires an in-depth evaluation and a discussion that compares these results and this method to other studies on changes in sea ice cover.

The discussion subsequently doesn't focus on the derived information from the HSIA (the trend line) but more on potential implications of the found changes for the people in those Alaska communities. These implications are based on a literature review, which makes the manuscript two sided and overcharged in information variety. Further the BSI is introduced too late and only covers a short part of the study which poses the question if it is really needed or useful.

In summary, the paper in its current state is unfocused and lacks detail in key sections.

**General Questions:**

**Does the paper present novel concepts, ideas, tools, or data?**

Yes, using the HSIA a historical record for the break-up and freeze-up of sea ice for four Alaska communities is presented. However, the literature review (mainly in the discussion section) does not present new findings.

**Are substantial conclusions reached?**

No

**Are the scientific methods and assumptions valid and clearly outlined?**

Not fully, since the methods section lacks a statistical evaluation and later sections (results and discussion) further highlight methodological concepts that were not introduced nor discussed.

**Are the results sufficient to support the interpretations and conclusions?**

Not fully, since the derived trend line is not statistically evaluated.

**Is the description of experiments and calculations sufficiently complete and precise to allow their reproduction by fellow scientists (traceability of results)?**

Not at all.

**Do the authors give proper credit to related work and clearly indicate their own new/original contribution?**

Yes

**Does the title clearly reflect the contents of the paper?**

No, the title mainly focuses on one aspect of the article (literature review in discussion). A better title would include the derived estimate for a change in the date of freeze-up and break-up. For example, it could be: 60 years of historical ice cover data reveal a significant shift towards a longer open-water season.

| |
|---|
| **Does the abstract provide a concise and complete summary?** |
| Yes, although there is potential for improvement. |
| **Is the overall presentation well-structured and clear?** |
| Partly; owing to the fact that the article presents a mix of a data analysis with a literature review concerning potential impacts for the local communities. A comparison of the changes in sea ice with actual impacts for the communities or potential impacts for four different communities would have been interesting but the impacts are all discussed in a very general and theoretica/hypothetical way. The discussion section is unnecessarily long and does not focus on the actual work done by the authors. Furthermore, the figures are of low quality. |
| **Is the language fluent and precise?** |
| Yes |
| **Are mathematical formulae, symbols, abbreviations, and units correctly defined and used?** |
| Mostly |
| **Should any parts of the paper (text, formulae, figures, tables) be clarified, reduced, combined, or eliminated?** |
| Yes, to all of the abovementioned sections. |
| **Are the number and quality of references appropriate?** |
| There are no major references to the methodological part of the paper and almost no references to other studies on sea ice cover. The literature review on the other hand seems to have a good basis of literature. |
| **Is the amount and quality of supplementary material appropriate?** |
| NA |

**Major review points**

**Abstract**
▪ The term "subsistence hunting" could be introduced once, and then it should be clear that the indigenous people of Alaska rely on the availability of game for their food supply, henceforth it would not be required to always indicate it again (also true for the whole article).
▪ Maybe the section that explains the HSIA is not necessary (defining it here as a historical atlas of sea ice cover would be enough).

**Introduction**
▪ P.1, L.14 and 17: If the focus is on food security (rather than the impact on coastal communities in general), why is the first example given related to soil erosion? Soil erosion is likely not the main impact on food security.
▪ P.1, L.15: the reader is introduced to the terms "direct" and "indirect impacts". However, for the indirect impacts, the term "globally-induced impacts" is also used. In the conclusion again the term indirect impacts is used. We suggest to use the same term in throughout the manuscript.
▪ P.1, L.21: Why not cite peer-reviewed literature?
▪ P.2, L.3: It is not quite clear what the term "place-based nature of climate change impacts" refers to. A short description or explanation would help.

**Methods**
▪ P.2, L.27: the term "best analog representations" was used. What does this mean? For a better understanding of the HSIA it is crucial to know what is meant with "analogs".
▪ For this study four communities and one offshore area were chosen. The manuscript states that the communities were chosen because of their "wide range of sea ice regimes, with varying levels of dependence on subsistence activities […]". For a better understanding of the communities it would be useful to have a short site description for all of them including the

reason why a particular location was chosen and further reference to the map in figure 1. Also, it is not clear why this particular location in the Bearing Strait location was chosen.

- P.3, L.3: it would be useful to support this claim with a reference to a date for which satellite data would be available (also add a source for the date).
- P.3, L.10: first a concern is raised that the data is heterogenous, and then it is only partly explained why and how the HSIA is nevertheless a good source. How did you test if there was an anomalous discontinuity?
- P.3, L.16ff: it is not clear how the area was selected, was it done manually? How was the calculation then conducted? Overall this section lacks detail and the results are not reproducible.
- P.3, L.24: the reasoning to why the 30% threshold was chosen is not clear, if 15% does the same as 30%, then why chose 30%? What did other people do to evaluate freeze-up and break-up of ice cover from gridded data?
- Overall: No information on statistical analyses that were used is given. Figure 2 / 3 hint at the use for linear regression, but that is the only information the reader gets from the article. The low "% variance explained" suggests that these trendlines may not be significant? Did you do a statistical test to determine if the trendlines are significant? Also, why was a linear trendline chosen? Does a stepwise (two-part) regression fit the data better?

**Results**

- Generally, this section relies heavily on the linear trend, although the linear trend is not mentioned in the methods nor is its quality assessed.
- It would be good to have a table (similar to Table 1) with all important information and statistical measures (not just the % of explained variance).
- The presentation of the Results is incomplete:
  - The studied community "Nome" is only mentioned in the Methods. Is there a reason why it is not presented in the Results and Discussion?
  - In Figure 4 suddenly "Wales" appears, without introducing it before.
  - Figure 2 and Figure 3 do not contain the results of "Nome" and "Bering Strait". After the introduction of the four communities and one offshore area in the methods, it is necessary to show all results or at least state why something is not shown / presented.
- P.4, L.9-15: This focuses on the explained variance by the linear trend, however it is just a qualitative description and raises more questions than it answers → no statistical evaluation!
- In section 3.2, lines 21-24 are already interpretation and should be moved to the discussion.
- Figures 2 and 3: The data for Kotzebue flips back and forth between two values in the 1960s and early 70s. Is this a data limitation issue?
- Figure 4: top 1%, means that for each period a different threshold is chosen. That makes comparisons of the different periods difficult. Also what are the methods used to determine the number of storms? Add a reference.

**Discussion**

- The quality of the produced data is never questioned nor is it assessed. How robust are the results?
- P4L26: These are at most potential impacts. Unfortunately, the real impacts are never determined or analysed. So change title to reflect that these are literature based potential impacts.
- P.5, L.5ff: the manuscript refers to Kotzebue Sound which "shows less of a change in freeze-up and breakup trends". This could be because it is surrounded by land on three sides. But what about the community of Nome which is on a similar, but not so distinct, location and shows the same trend in figure 4?
- P5.L34: Is there any evidence for that? Any data? any references?
- P7L21: If this has been reported, add the reference.

- P8.L3: This statement should probably come much earlier.
- P.8 (section 5.1): the manuscript starts again with an introductory part, then a description of the methods, results and discussion. For a clear structure however, a separation into the correct chapters would be necessary. Within this section the Barnett Severity Index (BSI) is very briefly introduced. Here the reader should get more information about the BSI. What is it? Why is it used? The BSI is only mentioned again in the first paragraph of the conclusions and is thus a minor part within the manuscript. Thus, one should introduce it properly and then also show its importance.
- The discussion has a major focus on the impacts. However, there are other aspects as for example species extinction that could be more dramatic with more open water days. Further the protection of species in danger from extinction could lead to conflicts with traditional hunting.

**Conclusions**
- P.10.L7-9: We agree with this but what is the relation between this statement and your actual study/analyses? The four different sites are not discussed in detail and it remains unclear how different these communities are or how different the impact of climate change and changes in sea ice has been for these communities.
- P.10, L.13: the usefulness of the BSI is described. This statement would be more logically placed together with the rest of the BSI, currently in section 5.1.
- So far, the direct impacts were always presented before the indirect impacts. On page 10, line 25ff the order was changed which is misleading. Always keep the same order.
- Most of the conclusion is actually a discussion and not a conclusion from the analyses presented in this study.

**Minor points**
- Some sentences are hard to read, e.g. P2L3-4, L20-21.
- P.2, L.29: Sea ice concentration = sea ice cover. Introduce this definition already in the introduction.
- Figure 1: Create a more useful and visually more attractive map. This map looks like it comes out of a video game of the 80s.
- The legends in figure 2, figure 3 are too small and do not contain the necessary information.
- Figure 5: don't just copy from Chapman, replot using the same style as the other plots. In general, all plots should have the same style, so also figure 4 needs to be adapted.
- Figure 8: add the 1:1 line
- Table 1: maybe it is better to give the trend in days per decade so that it is not a fraction of a day, which could suggest hourly data are needed
- Regarding the figures in general: no titles needed, clearer figure captions needed (a,b,c…). Also make sure to add informative legends.
- P.3, L.8: the word "of" is not needed
- P.4, L.21: replace "has" with "could have" because we cannot be sure about this.
- P.6, L.25: "polynas" → actually called "polynyas"
- P.8, L.12: the introduction of the BSI is suboptimal. It is here introduced as "severity of ice conditions index" whereas the acronym itself translates to "Barnet Severtity Index". Only after the first introduction of the correct terminology abbreviations should be used.
- P.8, L.24: also give the 5 nautical miles in kilometers (in general only use one distance measure)
- P.9, L.14: "will" or "can"?

---

## Referee Comment (RC3) · Anonymous Referee #3 · 1 Dec 2017

1. Does the paper address relevant scientific questions within the scope of TC? Yes 2. Does the paper present novel concepts, ideas, tools, or data? The ideas are important and novel, but are not well developed. 3. Are substantial conclusions reached? No. The paper draws conclusions (many of which may be inaccurate) from a weak analysis. 4. Are the scientific methods and assumptions valid and clearly outlined? No 5. Are the results sufficient to support the interpretations and conclusions? No 6. Is the description of experiments and calculations sufficiently complete and precise to allow their reproduction by fellow scientists (traceability of results)? No 7. Do the authors give proper credit to related work and clearly indicate their own new/original contribution? Yes 8. Does the title clearly reflect the contents of the paper? Yes, although I don't think the paper is well positioned to properly discuss or draw conclusions about

the full scope of impacts that they attempt to address. 9. Does the abstract provide a concise and complete summary? Somewhat 10. Is the overall presentation well structured and clear? Somewhat 11. Is the language fluent and precise? Yes 12. Are mathematical formulae, symbols, abbreviations, and units correctly defined and used? Not relevant. 13. Should any parts of the paper (text, formulae, figures, tables) be clarified, reduced, combined, or eliminated? The methodology section should be expanded to provide greater details on how the BSI was replicated using the HSIA data. 14. Are the number and quality of references appropriate? Yes 15. Is the amount and quality of supplementary material appropriate? Not applicable.

The paper relies on a limited and simplistic analysis to draw weak and poorly supported conclusions. The authors appear to acknowledge the limitations of their analysis, but precede to venture into topics far removed from their analysis. For example, the discussion of sea ice change on migratory marine mammals was quite disconnected from the analysis of local ice conditions near communities. Also, in terms of analyzing the "days left for whaling", the authors conclude that the spring ice-based whaling season has been cut in half (from 160-180 days to approx 80 days), when in reality the spring whaling season has never been much more than late-April through early June (<60days). (I comment more on this in my more detailed attached comments.) Little effort was made to address the simplification of their assumptions, although they point it out themselves in several cases (for example, by noting that their analysis is not sufficient to track the presence of landfast ice). As further example, the authors pointed out that their simple definition of "transition seasons" based on sea ice concentration thresholds may be problematic. I agree, and suggest that the authors think carefully about what new data and evidence they can introduce to this paper to make their quantitative results better provide a relevant context for the discussion on complex impacts to communities.

See my detailed comments in the attached PDF.

[Figure]

Please also note the supplement to this comment:
https://www.the-cryosphere-discuss.net/tc-2017-211/tc-2017-211-RC3-supplement.pdf
* * *
[Figure]

**Supplement:**

**DETAILED REVIEWER COMMENTS:**

Abstract: "reduced access to subsistence hunting species" should be reworded. Its not the species that are doing the hunting.

Abstract: The abstract doesn't seem to connect to the title. The title references "impacts" but the abstract describes that the main results pertain to summarizing sea ice trends.

Pg 1, Line 20: Suggest changing to "while changes in the seasonality and extent influence the migration of…"

Pg 2, Lines 3-6: The first two sentences are very vague and unclear. What are the challenges of the placed-based nature of climate change? The second sentence is confusing and seems to go off topic by referencing how research is to provide benefits. What potential benefits are you talking about?

Pg 2, Lines 11-13: While it may be true that the timing of break-up is more important than ice thickness to a local community, the authors should also consider that the definitions that scientists use to define break-up, which are determined in part by the observational methods and limitations, may also be very detached from how a community observes and defines break-up. Therefore, it is not only the variable that's important but also the definition and observation of that variable.

Pg 2, line 15: What is meant by rotten ice?

Pg 2, lines 16-17: It is not clear what is meant by "this type of metric". Also, I don't understand how the authors successfully made the argument that incorporating indigenous knowledge allows of the use of large-scale datasets to examine local impacts. Is it that local experts are able to embed their local observations within longer-term climate records?

Pg 2, Line 31: "with varying levels of dependence on subsistence activities, such as susceptibility to coastal erosion and interaction with the offshore oil and gas industry" How are susceptibility to coastal erosion and interaction with industry examples for dependence on subsistence? This sentence needs rewritten.

Figure 1: Does the grid cell in the Bering Strait overlap with the Diomede Islands? Since this is the only map in the paper, the paper will be improved by an improved map that shows the community locations in a bit more details.

Pg 3, Line 4: Please reference the passive microwave satellite record if that is in fact what you are referring to.

Pg 3, line 7: correct to "a different number of"

Pg 3, line 19: The paper would benefit by an expandED methodology. For example, it is not clear why " the maximum concentration…was extracted from within this area"? Why was the maximum extracted and not the average value. Was data analyzed across the entire annual cycle?

Pg 3, line 22: Instead of calling it freeze-up and break-up, I wonder if more accurate terms may be "ice-on" and "ice-off". I recognize that these are not necessarily common disciplinary terms, but since you are dealing with relatively small study areas, ice coverage can cross the threshold

quickly due to a shift in wind and thus may have nothing to do with a real phase change (which is implied by "freeze-up").

Pg 4, line 1-3: The methodology used to treat the years at Barrow when the ice coverage didn't drop below 30% is not clear to me. Please explain in greater detail how you where able to use the 45% or 60% threshold for these years, and integrate back into the longterm dataset. Also, I cannot easily see (too small) within the right-most panel in Figure 2 which years are when the ice never dropped below 30%.

Pg 4, line 2: This paragraph is about freeze-up yet it references "break-up dates".

Pg 4, line 7: The linear trend at Barrow for break-up was not calculated also. This should be stated.

Pg 4, Line 10: "Kotzebue shows 132% of the variance of freeze-up day for Shishmaref, and 108% the variance of break-up day." I understand what is being said here, but the wording needs to be clearer.

Figure 5: Please be clearer in the presentation. Red is the number of extreme storms during the open water period. Here, how and where is the open water period defined? Is this also using a 30% threshold?

Pg 5, Lines 26-34: This analysis of the "days left for whaling" is close to meaningless. With a nominal start date of April 15, 80 days, which is what is shown for recent years, would theoretically allow for whaling through the beginning of July. The bowhead hunt never really went too far past early June. It is true that the ideal ice conditions for ice-based spring hunting is being shortened, but these results do not reflect those trends. Looking at the earlier years, the authors show between 140-180 days for whaling, which would put the hunt all the way into fall, which doesn't make sense. This analysis seems to imply that ice is the only important piece to whaling. The authors acknowledge this somewhat by saying that "the end of whaling season does not necessarily coincide with the break-up of the landfast ice or the retreat of ice from the coast" and further note that their analysis may not capture the finer-scale resolution required to track landfast ice. This is an understatement. This analysis bears little relevance to landfast ice, and especially from the perspective of how a community may use landfast ice.

Pg 6, lines 8-12: This paragraph seems to overlook that the community of Utqiagvik is already adapting by hunting more in fall. This should be discussed.

Pg 6, line 24-27: These statements are not well-supported and may be inaccurate. Does Clarke's paper reference changes in hunting? I suspect not. The bowheads for the BCB stock begin to arrive in the Beaufort in late April/early May, not August. (Perhaps the authors are trying to say that bowheads migrating west from the eastern Beaufort are arriving to the western Beaufort near Pt. Barrow earlier in Fall?) Also, I am not sure there is any literature that shows that the bowheads passage through Bering Strait is tied to local ice conditions there. If there is, it should be referenced. This statement seems quite speculative.

Section 4.3: This entire section that discusses impacts on marine mammals, which rely on large regions and migrate through diverse ice conditions, seems disconnected from the results of this paper, which focus on local conditions near specific communities.

Pg 8, Line 8: The authors point out that their simple definition of transition seasons based on sea

ice concentration thresholds may be problematic. I agree, and suggest that the authors think carefully about what new data and evidence they can introduce to this paper to make their quantitative results better provide a relevant context for the discussion in this paper on impacts to communities.

Section 5.1: A great explanation of the methodology to recreate the BSI from the HSIA data should be included in the methodology section.

Figures 7 & 8: Why are the upper limits of the BSI shown in Figure 7 not represented in Figure 8 (e.g., values above 1000)? Does Figure 8 correspond to a subset of earlier years?

Pg 8, line 30: change to "are very likely"

Pg 9, line 20: Is there any evidence that increased shipping is leading to more goods, and a greater diversity of goods, to Arctic communities?

Pg 10, line 20: This conclusion regarding the traditional spring hunt being cut in half due to ice conditions is not accurate and is not well supported by the data presented in this paper. See my earlier comments. This conclusion, above all else in this paper, should not be published.

Pg 10, line 24: There is absolutely no evidence presented in this paper or relevant literature cited that indicates a change in bowhead whale migrations.

Pg 11, line 5: Where is the evidence that hunters are evaluating risk differently than in the past? The claim that hunters today walk on thinner ice than they used to because of the pressures of environmental change and hunting regulations seems over-simplified and perhaps altogether inaccurate.

---

## Author Comment (AC1) · 20 Dec 2017

**Reply to Referee #1**

**Impacts of a lengthening open water season on Alaskan coastal communities**

**Rolph *et al.* (2017), tc-2017-211**

We would like to thank Referee #1 for his/her constructive comments, which have helped improve the quality of the paper. We present below our detailed responses to the comments in blue.

1. Overview and major comments

This paper aims to use the Historical Sea Ice Atlas (HSIA) to assess potential direct and indirect impacts from sea ice change for 3 (or 4) Alaska communities. Unfortunately, the analysis described does not sufficiently support the paper's conclusions, due to a lack of focus and a clear connection to the primary analysis. The paper simply tries to cover too much ground. Further development, extension, and quality-control of the HSIA analysis is recommended, with additional explanation of the physical context of the sea ice environment at study locations, and more explicit linkages to community interaction and feedback regarding key impacts.

In order to provide more explicit linkages to community interaction and feedback regarding key impacts, we have introduced new indices that are based on community knowledge and experience, as reported in other studies. We have also drawn upon an additional data product (WRF-downscaled ERA-Interim winds) in the calculations of the new community-relevant indices. Specifically, the augmented set of indices includes: the number of 'false freeze-ups' (number of times ice concentration oscillated above and below the threshold value before true freeze-up was finally achieved), 'false break-ups', timing of freeze-up and break-up, the length of the open water duration, number of days where the winds are too strong to hunt via boat (wind speed thresholds from Ashjian (2010), and number of wind events capable of performing geomorphological work or damage to infrastructure from creation of waves and storm surge (using the definition of these wind events from Atkinson (2005) and Solomon (1994). By building our analysis around these metrics, we demonstrate how local and indigenous knowledge can inform use of a large-scale dataset to form locally-relevant indices describing change in sea ice conditions, which also leads to changes in the duration and strength of winds over open water.

Review metrics:

1. Does the paper address relevant scientific questions within the scope of TC?
Yes, but it is encumbered by side issues.
We feel that we have managed the side issues by refining the scope of the paper, but without more details about these side issues we cannot address this more specifically.

2. Does the paper present novel concepts, ideas, tools, or data?
Partly, but not fully realized.
We are not aware of any other study which has based a study on these indices for the communities examined in this study. We also feel that the newly added indices further demonstrate the value of a more generalized method to apply large-scale datasets to examine locally-relevant impacts of environmental change.

3. Are substantial conclusions reached?

No.

We have provided new conclusions about the recent decades (1979-2014) of the following indices for three communities: Number of false freeze-ups, number of false break-ups, number of open water days deemed too windy for offshore subsistence hunting, number of wind events capable of performing geomorphological work (erosion) or damage to infrastructure or habitats.  From 1953-2013, we have also presented changes in the timing of freeze-up and break-up for the communities of Kotzebue and Shishmaref. In Utqiagvik, there has been an approximate tripling of the number of wind events capable of significant coastline erosion from 1979-2014.  We believe that our documentation of changes in the various indices have led to new conclusions about environmental changes of dual relevance to Arctic coastal communities.

4. Are the scientific methods and assumptions valid and clearly outlined?

No – though if better explained and more thoroughly executed they may be.

We appreciate the reviewers comments and have expanded in the methods section.  Specifically, we have outlined how we have used each reference pertaining to the appropriate index, and explicitly how we have used the climate-related thresholds (e.g. sea ice concentration or wind speed) in the development of each index timeseries.

5. Are the results sufficient to support the interpretations and conclusions?

No – the discussion section relies too little on the results of the analysis.

The discussion has now been organized into subsections, each of which builds upon results from the previous subsections, leading to conclusions about marine access and coastal vulnerability.

6. Is the description of experiments and calculations sufficiently complete and precise to allow their reproduction by fellow scientists (traceability of results)?

Only in part. The method of recalculating the Barnett Severity Index (BSI) was not given.

Based on this and other review comments about the necessity of the BSI, we have removed the BSI (including its use and discussion) from the manuscript. Previously, we had cited Drobot (2003) for the methods of calculating the BSI.

7. Do the authors give proper credit to related work and clearly indicate their own new/original contribution?

Mostly.

Without further specification of which related work this pertains to, we cannot address this comment in detail, but we have made an effort to clearly outline (in Section 2) how our indices were developed by using appropriate references that have used indigenous knowledge, especially in the revised methods section.

8. Does the title clearly reflect the contents of the paper?

The title is nominally accurate but the paper does not succeed due to overreach.

Our title in our original manuscript is "Impacts of a lengthening open water season on Alaskan coastal communities." In our revision to the reviewers comments, we state how each of our indices relates directly to impacts on Alaskan coastal communities. Moreover, trends found in most of the indices are also directly related to the lengthening open water season.   We have decided to add a subtitle such that now our title reads "Impacts of a lengthening open water season on Alaskan coastal communities: deriving locally-relevant indices from large-scale datasets and community knowledge"

9. Does the abstract provide a concise and complete summary?
Improvement is needed.
Since we have changed the focus of the paper from analyzing direct and indirect impacts to the development of the locally-relevant indices, and have changed the abstract accordingly.

10. Is the overall presentation well-structured and clear?
No.
We appreciate this comment, and have made a major effort to re-organize the manuscript such that each index has a corresponding appropriate subsection. The subsections are linked throughout the Methods, Results, and Discussion main sections.

11. Is the language fluent and precise?
Improvement is needed, especially toward the use of plain language instead of jargon.
We have reviewed the wording for jargon, but would appreciate more detail in order to address this comment. For example, we have added a description of what 'rotten ice' is in the Introduction section (partially melted and weak). We have also made an effort to be descriptive of the added dataset (WRF-downscaled ERA-Interim) for readers who are unfamiliar with this dataset.

12. Are mathematical formulae, symbols, abbreviations, and units correctly defined and used?
Yes.

13. Should any parts of the paper (text, formulae, figures, tables) be clarified, reduced, combined, or eliminated?
The organization of the paper is poor, it attempts to cover too much ground on the basis of too little new analysis, and it would be improved by substantially editing the discussion section down to those questions clearly related to the HSIA analysis. The quality of the figures is low.
We have removed the BSI in response to the "attempts to cover too much ground" comment, and we have sharpened the focus on the user-relevant indices. As mentioned in our response to comment #10 above, we have reorganized the Methods, Results, and Discussion into their own subsections such that each index we have developed can be clearly followed from how it was calculated from the large-scale datasets of HSIA and newly-added WRF-downscaled ERA-Interim dataset. The quality of the figures has been addressed by replacing some of them.

14. Are the number and quality of references appropriate?
References for recent sea ice research papers are lacking; references to similar studies on indigenous knowledge of sea ice for the St. Lawrence Isl. communities, for example, are incomplete.
We have now restructured the paper such that the references citing community-based knowledge that we have used to develop thresholds are a central component of the paper and clearly described in the Introduction and Methods. References pertaining to walrus harvest and how open water impacts the sea ice conditions (accelerates the break-up of shore-fast ice), have been added to the restructured Discussion section concerning the transition period between ice and open water. For example, the reference Krupnik (2002) has been added there.

15. Is the amount and quality of supplementary material appropriate?
N/A

Abstract:

The abstract could be more concise with the removal of some unnecessary phrases like 'It is often remarked' (Pg1 L1) which only weakens the statement that 'Alaska coastal communities are on the frontlines...' This statement has been removed.  Also the phrase 'navigational regime shift' (Pg1 L11) is not the best choice given the freighted meaning of the term 'regime shift' in the physical sense, and further obscured by reference to 'navigational'. One could simply state that the impact of ice conditions on vessel operations near Utqiavik (Barrow) have eased since … [because]. The term 'regime shift' here and elsewhere is an example of jargon that doesn't add materially to the paper and should be avoided as it is not essential to the work at hand. The reviewers comment here refers to the shift seen in our extension of the BSI timeseries.  As noted earlier, we have followed recommendations of the other reviewers and removed the BSI in the new version of our manuscript.

Introduction:

It would be helpful to mention the role of the Barnett Severity Index (BSI) since it is highlighted in the abstract. It is also reasonable to provide the specific date range of the study, and a general sense of the application of "community feedback and interaction" with the research team, or, if indirect, the sources of indigenous knowledge incorporated in the study (i.e. collected interviews of elders, joint review of archived interviews, previous published work, etc.). It may also be appropriate to mention work by I. Krupnik, for example, that is of a similar nature. The introduction would also be a good place to mention the specific communities discussed in the paper (note that the community of Wales is introduced in results section 3.2 for the first time). For this paper an organization statement in the last paragraph of the introduction would be useful. We appreciate the comments here and have changed the introduction substantially.  As mentioned previously, the BSI has been removed from the new manuscript and so is no longer in the introduction. We have mentioned the work done by I. Krupnik (2003) which is highly relevant.  The introduction is also where we have now clearly outlined the work of Ashjian (2010), Atkinson (2005), and Solomon (1994) which have each identified thresholds using indigenous knowledge necessary for developing several of the indices now presented in this paper which did not exist previously (such as number of days 'too windy' for subsistence hunting via boat, and wind events capable of doing geomorphological work or damage to infrastructure). We have also added a statement near the end of the introduction, outlining the paper's contents and organization.

Data and Methods:

The structure and completeness of this section could be improved by separating and briefly addressing each element identified previously, for example: a) HISA and its underlying data and related issues/benefits for the specific period of the study (e.g. blend of surface and satellite obs, use of microwave data near the coast, impact of quarter-monthly resolution); b) sources of indigenous knowledge used in the study and how they were used; c) selection of study areas and reference HISA grid cell locations in light of (b) or otherwise; d) selection of metrics, as in 30% ice concentration threshold (Serreze et al. 2016 doi:10.1002/2016JC011977 provide a good rationale); e) BSI, which is absent in this section and the introduction but mentioned in the abstract – this is especially important as the BSI index uses sea-ice inputs only for the area north of Barrow and along the coast to Prudhoe Bay, hence its extension to distant study areas should be justified, and particularly given Barnett and Drobot's cautions regarding use of historical data prior to ∼1978 and the impacts of potential ice loss on the utility of the BSI; losses that have now taken place.  We have restructured the Data and Methods section as suggested by the reviewers comment here.  It is now organized into subsections, which describe the datasets used (HSIA and WRF-downscaled ERA-interim products), the methods for selecting grid cells representative of each community and the derivation of locally-informed indices The Serreze et al (2016) reference has been added as per the Reviewer's suggestion.

f) some brief general description of how direct and indirect impacts will be evaluated in light of results (e.g. erosion, travel for hunting, prey availability, transition season, as well as indirect ones) would be useful here. There are elements more appropriate for the introduction and/or discussion section included here that should be removed (e.g. line 13 and following). Since we have removed the BSI, lines 13 and following have also been removed. Based on this and the other reviewer comments on the need for a refined scope of this manuscript, we have shifted the focus of the paper from the differences between direct and indirect impacts, to the development of locally-relevant indices utilizing two large-scale datasets of HSIA and WRF-downscaled ERA-Interim. As such, a 'brief general description of how direct and indirect impacts' are evaluated is no longer relevant in this case.

Results:
The issue with Barrow in the first paragraph requiring an alternate calculation for break-up date should be explained in the methods section, and especially justifying the linear interpolation. It appears that in some years the ice never moved back sufficiently from the coast, and thus the interpolation leads to the inference of a condition that may have never physically existed. Inspection of the pertinent variables given by Barnett (1976) show 11 years (of 23 examples) 1953-1975 where the sea ice did not retreat at all (as measured in nautical miles), or not far enough to be detected in the reference grid box (esp. 1975). We have removed the linear calculation for Barrow because some years indexed did not have a true 'freeze-up and break-up event'. The number of false freeze-ups and false break-ups have been given for Barrow from the ERA-Interim dataset, as explained in the new version of the manuscript, and we feel this is a better representation of the change being seen there. Interpretation of these results depends on the physical context that helps explain why differences exist between the various study locations. For example, ice cover near Wales is influenced by the dynamics of the Bering Strait inflow, and local/advected solar heating in this and especially the Alaska Coastal Current (ACC). The ACC is not well observed by the Bering Strait mooring array (Woodgate 2012 and related papers) and should probably be considered by reference to additional sources. The Kotzebue area is dominated by fast ice, influenced by local solar heating, river discharge plumes and related freshwater stratification, and factors related to the ACC at the outer edges, and is probably less sensitive to dynamics (i.e. ocean currents, wind-driven flaw leads, etc.) until later in the break-up process, compared to other locations. The amount of open water in Barrow, until recent decades, was defined primarily by the width of the lead between the coast/fast ice edge and the heavy polar ice pack (e.g. Hunt & Naske 1977), which is in turn influenced by (among other factors) large-scale atmospheric patterns such as identified by Barnett (1976) and others since. An important factor at Barrow then may be the major loss of multiyear ice in the recent period, and accompanying changes in the mobility and strength of the ice pack. This is perhaps the most important distinction between Barrow and other locations where first-year ice has been the norm over the entire study period. The organization of the results section could be improved, perhaps by treating each phenomena as a sub-section, and by adding some analysis of the physical context required to understand why the observed differences may exist. This may well throw light on how communities' experiences and responses vary in the face of change to be discussed in following sections. Most sections may benefit from an opening remark explaining what follows and its organization (this is often done in the last paragraph of the introduction as well).We have responded to this comment by adding a detailed description in what we feel fits best into the Discussion subsection 'Increases in the number of windy days over open water and open water duration'. This description more clearly discusses the trends in the results section of the change in open water days, and includes the suggested comments about freshwater input from Kotzebue Sound and the loss of multiyear ice from Barrow. We have also added the reference (Section 4.2) of Ahlnas and Garrison (1984) which discusses how the warmest water during the summer in the Chukchi Sea occurs in Kotzebue Sound, and also the extent of the Alaska Coastal Current. We have added as the reviewer has rightly suggested organizational sentences to the start of each subsection, as well at the end of the Introduction.

Discussion:

The connections between the results of the analysis of HSIA data and the generally vague and unfocused discussion sections are unclear. Finding that there have been various changes in the timing of break-up, freeze-up, and open water duration is not novel in itself, though the impact on communities may be, if better explained by the specifics of the HSIA analysis here and its linkage to indigenous knowledge. We understand this could be interpreted as unclear, and feel that the addition of the locally-relevant indices in their own subsections (4.1, 4.2, and 4.3) makes for both a more connected and clearer analysis/discussion of the results. At points in the discussion the authors undermine their purpose with statements like: "Comments like this highlight the complexity of the transitions between open water and ice-covered seasons. It therefore might be problematic to simply use a sea ice concentration threshold to define the shoulders of the break-up and freeze-up transition seasons. We examined variability in the duration of the 0-30% period, but found no significant trends" (Pg 8 L1). This is rather discouraging to the reader. This statement has been removed in the discussion. We believe that the number of 'false freeze-ups' (number of times ice concentration oscillated above and below the threshold value before freeze-up was finally achieved) and 'false break-ups' compared between the communities examined provides a more robust representation of the changes happening in each community during the transition season. The BSI is introduced in discussion section 5, where the authors describe extending the index from 2000 to 2013, but do not explain specifically how or why (which should have been done in data and methods if this was an important component of the paper). Then they report that the original index and the extended index (HSIA BSI) don't agree very well and this may be due to basic incompatibility of the input data. The BSI is not mentioned again until one sentence in the conclusion, which begs the question of why it was in the paper at all. It would be interesting to know if the BSI is still used for long-range sea ice forecasting (its original purpose) or for other reasons, but its inclusion in this paper is not warranted. We agree with the reviewer, and the inclusion of the BSI is not necessary for the main points of this paper. We are able to demonstrate, without the BSI material, the use of local information to guide analysis of temporal variability based on large-scale datasets. The trend of the BSI also matches the number of open water days in Utqiagvik well, and since the latter has remained in the paper, we feel this is another reason the BSI should be removed for clarity and conciseness.

Conclusions:

Relationships between the specific HSIA-based sea ice analysis reported here and impacts of a lengthening open water season on Alaskan coastal communities are not clearly shown, especially with respect to community interaction and feedback.

Based on the comments of this reviewer and the other reviewers, we have shifted the focus of the paper to the use of metrics informed by community knowledge to assess local impacts of sea ice change using larger-scale datasets, as outlined in our response to "Overview and Main Comments" above. When viewed in the context of the temporal variation documented here, this can give meaning to how a changing climate is impacting Alaska coastal communities. We have changed the Conclusion section to reflect this, where we summarize our findings on the changes in these indices for each community examined.

---

## Author Comment (AC2) · 20 Dec 2017

**Reply to Referee #2: Interactive and Supplementary Comments**

**Impacts of a lengthening open water season on Alaskan coastal communities**

**Rolph *et al.* (2017), tc-2017-211**

We would like to thank Referee #2 for her constructive comments, which have helped improve the quality of the paper.  We present below our detailed responses to the comments in green.

G. Ljubicic (Referee)
gitaljubicic@cunet.carleton.ca

**Interactive comments:**
1. Does the paper address relevant scientific questions within the scope of TC? Yes.
2. Does the paper present novel concepts, ideas, tools, or data? Yes.
3. Are substantial conclusions reached? They need to be better supported.
Based on the reviewer's comments, we have restructured the paper to emphasize the  results and discussion most directly relevant to the main conclusion: community-based metrics of sea ice have changed in recent decades.  Please see our response to Reviewer #1, Question 3:  We have provided a timeseries of the following indices for three communities from 1979-2014: Number of freeze-up/break-up cycles, number of open water days deemed too windy for offshore subsistence hunting, number of wind events capable of performing geomorphological work (erosion), or damage to infrastructure or habitats.  From 1953-2013, we have also presented changes in the timing of freeze-up and break-up for the communities of Kotzebue and Shishmaref. In Utqiagvik, there has been an approximate tripling of the number of wind events capable of significant coastline erosion from 1979-2014.  The sum of false freeze-ups and false break-ups have increased in recent years for all communities examined, especially Shishmaref. The number of days considered too windy for hunting via boat is increasing along with the increasing open water period for Utqiagvik, and maintains a significant fraction of the number of open water days in both Kotzebue and Shishmaref.  These conclusions are community-relevant because we draw our thresholds from studies which have used indigenous knowledge to develop thresholds of climate related variables we were able to analyze from our large-scale datasets.  We have also added the new dataset for wind, WRF-downscaled ERA Interim, as explained in our new manuscript.  This dataset enabled us to extend the study to include the number of days considered too windy for hunting via boat see sample plots below).  The threshold for wind speed used is 6 m/s over open water, as reported by whalers to hinder whaling success due to high waves (Ashjian et al, 2010).

[Figure]

4. Are the scientific methods and assumptions valid and clearly outlined? See comments
in attached .pdf.

Our responses to the reviewers comments here are given below in the responses to the supplementary
material.

5. Are the results sufficient to support the interpretations and conclusions? See com-
ments in attached .pdf.

Our responses to the reviewers comments here are given below in the responses to the supplementary
material.

6. Is the description of experiments and calculations sufficiently complete and precise to
allow their reproduction by fellow scientists (traceability of results)? In relation to HSIA
analysis, yes. In others, need to be better supported.

We are assuming here based on the other comments that the methods of calculating the BSI index was
not fully explained, and in the new manuscript, the BSI index has been removed.  We decided to do this
as well based on the concerns by the reviewer comments that it is not well connected to the rest of the
paper.  We also feel that the trends shown in the BSI index are captured, at least qualitatively, by the
changes in the number of open water days at Utqiagvik, which is still included in the paper.  The
methods for calculating the number of open water days is given clearly in the Methods subsection
"Selection of metrics to determine open water duration, freeze-up and break-up dates".

7. Do the authors give proper credit to related work and clearly indicate their own
new/original contribution? Yes, but could be better referenced.

We appreciate the reviewer's comments here, and have added more references to the paper to support
assertions made in the text and appropriately credit work of others.   Some examples of the added
references include work done by I. Krupnik (2002), Serreze  et al (2016), and the references from
which we have taken the climate-related thresholds for wind speeds over open water, Ashjian et al
(2010), Solomon et al (1994), and Atkinson (2005).

8. Does the title clearly reflect the contents of the paper? Yes.

9. Does the abstract provide a concise and complete summary? Yes.

10. Is the overall presentation well structured and clear? Yes, but needs a bit more context
in places.

In response to the reviewer's comments, we have restructured the presentation so that there is a
consistent framework of subsections (for the different metrics) in the Methods, Results, and Discussion
sections.  We feel this has resulted in a more clearly organized manuscript.

11. Is the language fluent and precise? Yes.

12. Are mathematical formulae, symbols, abbreviations, and units correctly defined and used? N/A

13. Should any parts of the paper (text, formulae, figures, tables) be clarified, reduced, combined, or eliminated? See comments in attached .pdf.

Our responses to the reviewers comments here are given below in the responses to the supplementary material.

14. Are the number and quality of references appropriate? See comments in attached .pdf.

Our responses to the reviewers comments here are given below in the responses to the supplementary material.

15. Is the amount and quality of supplementary material appropriate? Yes

We appreciate the reviewer's detailed comments in the supplementary comments provided below, and have addressed those as follows (responses in green).

**Supplementary comments:**

**November 30, 2017**

The premise of this paper is to highlight the impacts of a lengthening open water season on Alaska coastal communities. Overall I find it an interesting and highly relevant paper. There is much current interest in defining impacts of environmental change according to indicators of relevance in a community context, and thus ultimately to help support decision-making at different scales. The extended record of the Historical Sea Ice Atlas (HSIA) is a valuable dataset for analyzing trends over time, although there are multiple challenges in doing so at appropriate scales and within particular community contexts. This paper tries to cover a lot of ground in a short paper. There are a number of good points made, although several areas where I would like more clarification, appropriate reference support, and nuanced discussion. It is also a paper well suited for discussion, and I look forward to reading feedback from other reviewers and discussants. Below are my contributions to this iterative review process, organized according to the key areas where I feel revisions would be needed prior to acceptance for full publication.

We particularly appreciate the reviewers overall comments here, especially the comment about there being interest in "defining impacts of environmental change according to indicators of relevance in a community context...". We have restructured the paper so that now multiple such indices are included and think this sharpens the main purpose of the paper: to demonstrate how utilization of large-scale datasets can be informed by indigenous knowledge in development of products that can inform stakeholders. In this case, the products of this paper are the long-term timeseries of our indices. The challenges of using the datasets at appropriate scales has been addressed here by adding the use of the references which have developed the thresholds informed by community stakeholders. An example of such a threshold is the wind speed threshold of lower than 6 m/s over open water for successful subsistence hunting and safe travel via boat (Ashjian et al (2010)).

**1) Community uses of sea ice** – Given the premise of the paper to investigate the direct and indirect impacts of a lengthening open water season on four Alaskan coastal communities (Barrow, Kotzebue, Shishmaref, and Nome), I think it would be important to have more characterization of geographic (e.g. physical conditions, typical sea ice extent and cycles) and cultural (e.g. uses of sea ice particular kinds of sea ice for particular hunting or harvesting practices, links to seasonal traditions or community events) context up front. It could be added to the introduction, or be in a new "community context" section, but without this it makes it hard to interpret some of the arguments being made later in the Discussion. The references to communities are highly generalized, without much sense of how their uses or priorities for sea ice may be shared or different, and this would help to strengthen arguments

as well as deepen the relevance of the analysis to the communities in question. Related to this, there is little explanation for the selection of the four communities beyond the diversity of sea ice regimes and subsistence activities (but these are not really introduced). In addition, Nome does not appear in most trend analyses, and Wales appears inconsistently in the text. More explanation and consistency in the communities of interest would be important. Furthermore, many of the references I am used to seeing that describe community use, conditions, and importance of sea ice from local perspectives in Alaska could be better incorporated throughout this paper to support both the local context as well as the analysis of direct and indirect impacts (e.g. work by Huntington, Eicken, Krupnik, Norton, George, Druckenmiller, among others).

We agree there was inconsistency in the extent at which different communities were examined in the previous paper. Based on the reviewer comments, we have decided to focus on the three communities: Kotzebue, Shishmaref, and Utqiagvik. Also based on the suggestion of the reviewer, we have added a subsection entitled "Characterization of communities examined" to the introduction providing characterization of these three communities. This new subsection provides context and sets up the Discussion section for the rest of the paper. It includes geographic position, typical sea ice cycles, and how the sea ice is used seasonally for traditional subsistence hunting practices. For example, a reference for the case of Utqiagvik is Gearheard et al (2006) "It's not that simple": a collaborative comparison of sea ice environments, their uses, observed changes, and adaptations in barrow, Alaska, USA, and Clyde River, Nunavut, Canada. *AMBIO: A Journal of the Human Environment, 35*(4), 203-211.

**2) Selection of 30% ice concentration threshold –** The choice of selecting 30% threshold for freeze-up and break-up needs more discussion and justification, as well as greater consideration of associated limitations. I find this to be quite low, if considering travel on landfast ice. It would also be very helpful to more clearly relate this to community use of sea ice. What would local perceptions of freeze-up approximate to in terms of ice concentration? You cited some of my previous work (Laidler et al., 2009) in which we used 9/10 (90%) ice concentration for freeze-up in terms of being navigable on snowmobile or foot (vs. 5/10 which is the common definition for freeze-up in relation ship navigation). At 30% concentration I would think there is still a lot of broken moving ice. Perhaps the overall trends would not change much, but this threshold selection is critical in terms of the arguments being made, and how this would translate to impacts on communities. This also has important implications for how transitional stages are considered, which are not really captured with one threshold (e.g. 30% used as break-up and as ice-free definition within the paper). I think this threshold selection and representation of transitional seasons is deserving of more careful consideration and/or articulation.

The purpose of our timeseries of freeze-up and break-up dates is to show the change or trend in freeze-up and break-up, which is, for the purposes of comparison across communities while using large-scale data sources, the date at which the sea ice concentration passes a particular threshold. Also in response to Reviewer #1, in their comments about section "Data and Methods", we have also re-visited the explanation for choosing the 30% sea ice concentration threshold, citing Serreze et al (2016). While we agree that 90% is a more suitable threshold for over-ice transport, the 30% threshold is arguably a compromise between the more commonly used 15% threshold for "ice extent" and a threshold for on-ice travel. In addition, the dates for various thresholds are well correlated and so the observed trends are not particularly sensitive to the choice of threshold value.

**3) Figure 1 –** This figure does not give a good sense of scale of ice area covered around each community, or resolution of grid cells. Could a larger and more detailed figure be created to better represent this?

Because the ice coverage varies so widely over seasonal (and interannual) timescales, we believe that a depiction of the sample grid cells to illustrate the resolution provides the most "bang for the buck". In the present version of Fig. 1, we show grid cells for the major communities included in the study.

**4) Interview citations –** Two interviews are cited in the text, and referenced as being interviewed in Kotzebue in 2013. There is no other context about these interviews in terms of how they were related to this research or other community-based projects, or any details in the Methods section about how interviews were undertaken and with what focus and which participants. I would like to see more of these local and Indigenous perspectives included in the paper, but they also need to be clearly explained and included in methods. Furthermore, interview quotes included from other papers need to be fully cited to the paper they were published in (as well as the individual), so they can be appropriately credited and contextualized.

We agree with the reviewer that this was not properly contextualized in the first version, and have made appropriate changes relating to both the citations as well as a clearer outline of the specific methods about how the indigenous knowledge was used in to obtain the main findings of the paper. The interviews now cited in the text are clearly credited with both the person being quoted and the source the quote appears in. The first interview comes from Ross Schaeffer, who was interviewed by Sarah Betcher in 2013. This is second interview quotation is by a hunter in St Lawrence Island, cited from work done by I. Krupnik (2002). The context for these interviews is more clearly given in the Discussion section. The quotation by Ross Schaeffer describes how the transition period seems longer in more recent years, and this is located in the revised Discussion subsection (4.1) which includes the changes in the number of false freeze-ups and false break-ups. The second interview is also located in this edited subsection, which describes how the increasing amount of open water can lead to a shorter period of break-up. In terms of the reviewers comment that she would like to see more local and Indigenous perspectives added in the paper, this is also included in the newly-added reference Ashjian et al (2010) because the authors of that study conducted interviews with local people in order to develop the wind speed thresholds we chose to use for our analysis.

**5) BSI interpretations –** This analysis does not seem well connected to the rest of the paper, and the calculations and methods involved are presented in the Discussion rather than Methods. Perhaps getting into this analysis in sufficient depth is beyond the scope of the paper? It would be good to really clarify what the primary goal and emphasis of the paper is. If it is indeed on community impacts (and related to community priorities and concerns), then expanding in areas noted above may be preferred to this particular aspect of analysis.

Based on this comment and the other reviewers comments, we have decided to remove the BSI analysis from the paper. Also see the comment by Reviewer #1 in the section pertaining to the Discussion.

**6) Societal levels and accessibility arguments –** I think what you are trying to refer to here is not societal levels (or scales), but decision-making or jurisdictional scales. This needs to be clarified throughout. The arguments here are also covered in such a generalized fashion, that it is difficult to connect to the sea ice and community-specific trend analysis. What would this mean in different community contexts? And when you talk about the accessibility of HSIA, to whom are you referring? How accessible and useable (and/or currently used) is the HSIA in Alaskan coastal communities?

We agree this section is not very well connected to the sea ice and community-specific trend analysis, especially in light of how the new manuscript has developed on the focus of the analysis of the indices

already based on thresholds from references using community-level input. Based on the extensive revisions to the updated manuscript, we feel this section did not really fit well with the flow for the rest of the paper, so it has been removed.

**7) Typos and References –** There are a number of minor typographic errors throughout, as well as a number of incomplete references, that need to be attended to. I can provide more details on these if requested.

We have revised the draft for typos, and checked for incomplete references. We would be interested to know which ones these are, in case something was missed that we are not aware of.  Some examples of the more complete references include better citation of the interviews, as discussed in Comment #4 above.

In the process of trying to compile my feedback, a number of other questions have arisen for me. But I will leave it here to see what the other reviewers and discussants say, and how the authors choose to respond. I am then happy to continue being part of the iterative review and discussion process.

We sincerely appreciate the detailed feedback which we feel has significantly improved the revised manuscript and are open to additional feedback.

Best wishes,
Gita Ljubicic
Department of Geography and Enviornmental Studies
Carleton University, Ottawa, Canada

---

## Author Comment (AC3) · 20 Dec 2017

**Reply to Referee #3: Interactive and Supplementary Comments**

**Impacts of a lengthening open water season on Alaskan coastal communities**

**Rolph *et al.* (2017), tc-2017-211**

We would like to thank Referee #3 for his/her constructive comments, which have helped improve the quality of the paper.  We present below our detailed responses to the comments in orange.

**Interactive comments**
**Anonymous Referee #3**

1. Does the paper address relevant scientific questions within the scope of TC?
Yes
We are glad to hear the reviewer agrees that the scientific questions raised in this paper are within the scope of TC.

2. Does the paper present novel concepts, ideas, tools, or data?
The ideas are important and novel, but are not well developed.
Please see the responses to Reviewers #1, Question 2, and #2, Question 3.  In summary, we have attempted to provide a more unified framework by including an additional variable (wind) to develop community-informed metrics that were not in the original submission.  The newly added indices include the number false freeze-ups and break-ups during each seasonal transition, the number of geomorphologically-significant wind events over open water, fractions of days deemed 'too windy' for subsistence hunting via boat, freeze-up/break-up timing, and open water duration provide methods. The methods and indices developed here can be applied not just to the three Alaska communities examined in this study, but also anywhere else along other coastlines affected by sea ice.  This applicability of our methods developed here to other places we feel is an important tool to share with the Cryosphere community.

3. Are substantial conclusions reached?
No. The paper draws conclusions (many of which may be inaccurate) from a weak analysis.
Please see the responses to Reviewer #1 and Reviewer #2, Question 3.  These responses demonstrate new results we have added to the manuscript which relate directly to our conclusions.  We have found that there is an increased number of combined false freeze-up/break-ups during the transitions between ice-covered and open water seasons in each community, as well as a significant increase of open water periods with winds too strong for subsistence hunters to hunt successfully via boat.  Our conclusions that the changing sea ice conditions (which are related to the change in windy conditions over the expanded open water period) are directly impacting Alaska coastal communities ability to maintain a subsistence lifestyle as well as (related) challenges they face in travel  over ice or water in the increasingly erratic transition season.

4. Are the scientific methods and assumptions valid and clearly outlined?
No
We appreciate the reviewers comments here, and accordingly we have added a new subsection for each method and assumptions used when developing each index, and feel this greatly improves the organization in this context. Please see also our response to Reviewer #1, Question 4, and our supplementary responses to Reviewer #2.  In summary, we have outlined how we have used each

reference pertaining to the appropriate index, and explicitly how we have used the climate-related thresholds (e.g. sea ice concentration or wind speed) in the development of each index timeseries.

5. Are the results sufficient to support the interpretations and conclusions?
No
Please see the example figure provided in our response to Reviewer #2 for Question #3. We have restructured the manuscript such that each result of our multiple indices calculated for each community has its own corresponding subsections in each the Results section and Discussion section. This improves the organization of the paper and allows for an easier-to-follow connection from the results to interpretation of the results.

6. Is the description of experiments and calculations sufficiently complete and precise to allow their reproduction by fellow scientists (traceability of results)?
No
Please see our responses to Reviewers #1 and 2, Question 6.

7. Do the authors give proper credit to related work and clearly indicate their own new/original contribution?
Yes
We are glad to hear this was clear in the manuscript and look forward to feedback if the reviewer feels this was not addressed for some reason in the new manuscript.

8. Does the title clearly reflect the contents of the paper?
Yes, although I don't think the paper is well positioned to properly discuss or draw conclusions about the full scope of impacts that they attempt to address.
Please see our response to Reviewer #1, Question 8: Our title is "Impacts of a lengthening open water season on Alaskan coastal communities: deriving locally-relevant indices from large-scale datasets and community knowledge ." In our updated version in response to the reviewers' comments, we chose to develop indices that relate directly to impacts on Alaskan coastal communities. The trends found in most of the indices are also directly related to the lengthening open water season, which we believe is a justification for including "open water season" in the title.

9. Does the abstract provide a concise and complete summary?
Somewhat
We have updated the abstract so that it concisely outlines our study communities, how we have developed each community-relevant index, which datasets were used, as well as the main conclusions from the changes seen in the results of the calculated indices timeseries.

10. Is the overall presentation well structured and clear?
Somewhat
Please see our responses to Reviewer #1 and #2, Question 10: We have taken much care to re-organize the manuscript such that each index has is organized into appropriate subsections. The subsections are then consistent across the methods, results, and discussion.

11. Is the language fluent and precise?
Yes.

12. Are mathematical formulae, symbols, abbreviations, and units correctly defined and used?
Not relevant.

13. Should any parts of the paper (text, formulae, figures, tables) be clarified, reduced, combined, or eliminated?
The methodology section should be expanded to provide greater details on how the BSI was replicated using the HSIA data.
We have expanded the methodology section such that the methods are organized into subsections about how we calculated each index. We have removed the BSI as we feel the conclusions drawn from results from the changes seen in the timeseries we extended can also be drawn from the newly-added indices in the updated manuscript. Please see also our responses for Reviewer #1, Question 13, and the supplementary section titled 'BSI interpretation' of Reviewer #2.

14. Are the number and quality of references appropriate?
Yes
We are glad to hear this, we have also added several references to the new manuscript as necessary based on the changes made.

15. Is the amount and quality of supplementary material appropriate?
Not applicable.

The paper relies on a limited and simplistic analysis to draw weak and poorly supported conclusions. The authors appear to acknowledge the limitations of their analysis, but precede to venture into topics far removed from their analysis. For example, the discussion of sea ice change on migratory marine mammals was quite disconnected from the analysis of local ice conditions near communities. Also, in terms of analyzing the "days left for whaling", the authors conclude that the spring ice-based whaling season has been cut in half (from 160-180 days to approx 80 days), when in reality the spring whaling season has never been much more than late-April through early June (<60days). (I comment more on this in my more detailed attached comments.) Little effort was made to address the simplification of their assumptions, although they point it out themselves in several cases (for example, by noting that their analysis is not sufficient to track the presence of landfast ice). As further example, the authors pointed out that their simple definition of "transition seasons" based on sea ice concentration thresholds may be problematic. I agree, and suggest that the authors think carefully about what new data and evidence they can introduce to this paper to make their quantitative results better provide a relevant context for the discussion on complex impacts to communities.
We have changed the manuscript substantially from the submitted version based on the reviewers' comments. Please see our responses to similar concerns expressed in the main comments of Reviewers #1 and #2. These are at the top of the response to Reviewer #1 titled 'Overview and major comments', and in the start of the supplementary comments for Reviewer #2. We also appreciate this reviewer's comment about the whaling figure. We address this also in the detailed comments below. What was meant in the original manuscript by the whaling figure is that, due to the earlier break-up timing offshore Utqiagvik, the spring whalers now have less time for hunting from the sea ice. We have decided to remove this figure from the manuscript due to this reviewer's comments. In its place, we now focus on the increase in the number of windy days seen offshore Utqiagvik which we speculate will impact the fall whale hunt (Figure of this index is given in response to Reviewer #2, Question 3, and is also present in the new manuscript). To determine what is deemed 'too windy' to hunt via boat, we have used the wind speed threshold from Ashjian et al (2010), who used interviews from a number of whalers from Utqiagvik. We have used the newly added dataset (WRF-downscaled ERA Interim (Bieniek et al, 2016) in order to obtain the wind data offshore of these communities. We calculated the number of days where the wind speed threshold from Ashjian et al (2010) was exceeded during the open water period and presented a timeseries of this as a fraction of 'usable' vs 'too windy' open water

days for each community. Pertaining to the Reviewer's comment about the transition seasons, we have shifted the focus from the duration of the the transition season to the number of times the ice froze-up and broke out before the 'true' freeze-up and break-up. We have provided an interpretation of our results for this index in its own subsection in the Discussion section of the new manuscript.
* * *
We appreciate the reviewer's detailed comments provided below. Our responses are in orange.

**Supplementary comments:**

Abstract: "reduced access to subsistence hunting species" should be reworded. Its not the species that are doing the hunting.
This statement has been removed based on the substantial changes we have made to the abstract in light of the newly developed indices described above and in the responses to the other reviewers.

Abstract: The abstract doesn't seem to connect to the title. The title references "impacts" but the abstract describes that the main results pertain to summarizing sea ice trends.
The abstract now clearly links some main results to the impacts. For example, in Utqiagvik, there has been an approximate tripling of the number of wind events capable of significant coastline erosion from 1979-2014, and also an increase in the number of days too windy to be hunting via boat.

Pg 1, Line 20: Suggest changing to "while changes in the seasonality and extent influence the migration of..."
This statement has been removed because it does not really fit with the new changes in the manuscript. However, we have added a similar statement in the newly added subsection "Characterization of the communities examined" where Shishmaref is described, stating that Shishmaref "is at the center of animal migration routes and also a center of a complex food-distribution system based in subsistence hunting practices Marino (2012)."

Pg 2, Lines 3-6: The first two sentences are very vague and unclear. What are the challenges of the placed-based nature of climate change? The second sentence is confusing and seems to go off topic by referencing how research is to provide benefits. What potential benefits are you talking about?
Key challenges of the place-based nature of climate change are that the impacts of climate are not felt equally across the globe, and that climate itself varies geographically around the Earth. For example, some places are getting drier while others are seeing more precipitation, and the associated impacts of these changes are what we meant by 'challenges of the place-based nature of climate change.' We agree that the second sentence can be seen as off-topic. In the new manuscript, both statements have actually been removed and the focus of that paragraph remains to explain the necessity of speaking with community members to understand what metrics are most important in the attempt at evaluating the impacts of climate change.

Pg 2, Lines 11-13: While it may be true that the timing of break-up is more important than ice thickness to a local community, the authors should also consider that the definitions that scientists use to define break-up, which are determined in part by the observational methods and limitations, may also be very detached from how a community observes and defines break-up. Therefore, it is not only the variable that's important but also the definition and observation of that variable.

We agree with the reviewers comment here, and have added an explanation to the added subsection in the Introduction titled "Identification of metrics useful for describing climate change-related impacts on Arctic coastal communities". We have cited Johnson and Eicken (2016) "Estimating Arctic sea-ice freeze-up and break-up from the satellite record: A comparison of different approaches in the Chukchi and Beaufort Seas". The explanation we added that, while freeze-up and break-up timing can vary based on data source, it is at the same time important to evaluate the timing of each in such a way that can be applied across communities. Using a sea ice concentration threshold is one way of doing this with available data.

Pg 2, line 15: What is meant by rotten ice?
Rotten ice in this context means weak and partially melted ice. This explanation has been added to the manuscript.

Pg 2, lines 16-17: It is not clear what is meant by "this type of metric". Also, I don't understand how the authors successfully made the argument that incorporating indigenous knowledge allows of the use of large-scale datasets to examine local impacts. Is it that local experts are able to embed their local observations within longer-term climate records?
By "this type of metric" we were referring to the way the ice breaks up, as discussed in the preceding sentence. In terms of how we have made the argument that "incorporating indigenous knowledge allows the use of large-scale datasets to examine local impacts", we have significantly extended this practice in the new manuscript by using thresholds identified by indigenous knowledge and applied that to our analysis of the large-scale datasets. This is described in more detail in our responses to Reviewer #1 titled 'Overview and major comments', and in our responses at the start of the supplementary comments for Reviewer #2.

Pg 2, Line 31: "with varying levels of dependence on subsistence activities, such as susceptibility to coastal erosion and interaction with the offshore oil and gas industry" How are susceptibility to coastal erosion and interaction with industry examples for dependence on subsistence? This sentence needs rewritten.
We agree with the reviewer comment that this sentence is unclear. This statement now exists in the subsection added to the methods section, which has been added based on the comments by the other reviewers. We have reworded it to "with varying levels of dependence on subsistence activities,  susceptibility to coastal erosion, and interaction with the offshore oil and gas industry."

Figure 1: Does the grid cell in the Bering Strait overlap with the Diomede Islands? Since this is the only map in the paper, the paper will be improved by an improved map that shows the community locations in a bit more details.
The grid cell in the Bering Strait is intended merely to show the resolution of the Historical Sea Ice Atlas in the offshore region. We included on the map the communities on which the present paper focuses.

Pg 3, Line 4: Please reference the passive microwave satellite record if that is in fact what you are referring to.
We have changed the phrase "satellite-derived data" to "passive microwave satellite record". This statement now exists in the reorganized subsection of the Methods labelled "The Historical Sea Ice Atlas".

Pg 3, line 7: correct to "a different number of"
Corrected to as suggested.

Pg 3, line 19: The paper would benefit by an expanded methodology. For example, it is not clear why " the maximum concentration...was extracted from within this area"? Why was the maximum extracted and not the average value. Was data analyzed across the entire annual cycle?

To answer the reviewer's comment we have added to this section: "The maximum concentration was extracted an not the average because if one grid cell in this area was higher than the others, it could serve somewhat as a 'choke point' or hazard while the other grid cells do not.  In order to capture this, we decided to take the maximum value of these several grid cells, although the neighboring grid cells do not typically vary significantly in concentration. The entire sea ice cycle was examined."

Pg 3, line 22: Instead of calling it freeze-up and break-up, I wonder if more accurate terms may be "ice-on" and "ice-off". I recognize that these are not necessarily common disciplinary terms, but since you are dealing with relatively small study areas, ice coverage can cross the threshold quickly due to a shift in wind and thus may have nothing to do with a real phase change (which is implied by "freeze-up").

We appreciate the reviewer's perspective here, but feel that freeze-up does not necessarily imply phase change in this context. We have added the following statement to the manuscript in the newly-added subsection of the Methods titled "Indices related to freeze-up, break-up, and duration of open water period" in order make this aspect more clear: "The timing at which freeze-up and break-up concentration thresholds are crossed does not necessarily imply a phase change, but also can include advection of ice in terms of shifts in winds or currents."

Pg 4, line 1-3: The methodology used to treat the years at Barrow when the ice coverage didn't drop below 30% is not clear to me. Please explain in greater detail how you where able to use the 45% or 60% threshold for these years, and integrate back into the longterm dataset. Also, I cannot easily see (too small) within the right-most panel in Figure 2 which years are when the ice never dropped below 30%.

Based on this comment and the other reviewer comments (see also Reviewer #1 comment about the Results section), we have decided to remove the timeseries of break-up and freeze-up timing for Utqiagvik.  The open water duration is still covered for Utqiagvik (see Figure in response to Reviewer #2 comment in Question #3 of Interactive comments).  Also, due to the significant changes in most of the indices evaluated in the new draft, we now have incorporated a separate timeseries figure for Utqiagvik given below.  This shows the increase in the number of open water days, along with the number of days deemed too windy for a successful whale hunt (wind speed thresholds taken from Ashjian et al (2010).

[Figure]

Pg 4, line 2: This paragraph is about freeze-up yet it references "break-up dates".
We have rearranged the organization of the paper based on the comments of the reviewers, and have combined explanations of the results pertaining to the freeze-up and break-up timing with the new results found in the change in the number of false freeze-ups and false break-ups.

Pg 4, line 7: The linear trend at Barrow for break-up was not calculated also. This should be stated.
As mentioned in response to previous reviewer comments (for example our response to Reviewer #3's comment above about Pg 4, line 1-3), we have removed the freeze-up and break-up timing trends for Barrow.

Pg 4, Line 10: "Kotzebue shows 132% of the variance of freeze-up day for Shishmaref, and 108% the variance of break-up day." I understand what is being said here, but the wording needs to be clearer.
We appreciate the reviewers comment and have reworded to: "Kotzebue shows more variability in the timing of freeze-up than Shishmaref, with 132% of the variance of freeze-up timing for Shishmaref, and 108% the variance of break-up timing."

Figure 5: Please be clearer in the presentation. Red is the number of extreme storms during the open water period. Here, how and where is the open water period defined? Is this also using a 30% threshold?
We have removed this figure because the impacts on the community are more thoroughly covered by the new indices present in the revised manuscript. The open water period was defined using a 15% threshold. These figures pertain to the number of days found to be too windy for hunting safely via boat, and also the changes seen in the number of wind events considered to generate significant enough waves to do geomorphological work (erosion), or damage to structure and habitats. The wind speed thresholds used for these were applied for open water, and come from the references of Ashjian et al (2010), Atksion (2005), and Solomon et al (1994). The change in days "too windy" for subsistence hunting via boat is given in the Figure in our response to Reviewer #2, Question 3. The change in the number of geomorphologically-significant wind events are given in the figure below, which is also included in the revised manuscript.

[Figure]

[Figure]

[Figure]

Pg 5, Lines 26-34: This analysis of the "days left for whaling" is close to meaningless. With a nominal start date of April 15, 80 days, which is what is shown for recent years, would theoretically allow for whaling through the beginning of July. The bowhead hunt never really went too far past early June. It is true that the ideal ice conditions for ice-based spring hunting is being shortened, but these results do not reflect those trends. Looking at the earlier years, the authors show between 140-180 days for whaling, which would put the hunt all the way into fall, which doesn't make sense. This analysis seems to imply that ice is the only important piece to whaling. The authors acknowledge this somewhat by saying that "the end of whaling season does not necessarily coincide with the break-up of the landfast ice or the retreat of ice from the coast" and further note that their analysis may not capture the finer-scale resolution required to track landfast ice. This is an understatement. This analysis bears little relevance to landfast ice, and especially from the perspective of how a community may use landfast ice.

We have removed this figure from the manuscript based on the reviewers comments.  As mentioned previously, what was meant in the original manuscript by the whaling figure is that due to the earlier break-up timing offshore Utqiagvik, the spring whalers now have less time for hunting from the sea ice. In its place, we have focused on the increase in the number of windy days seen offshore Utqiagvik which we speculate will impact the fall whale hunt (Figure of this index is given in response to Reviewer #2, Question 3, and is also present in the new manuscript).

Pg 6, lines 8-12: This paragraph seems to overlook that the community of Utqiagvik is already adapting by hunting more in fall. This should be discussed.

We at present cannot find a referenceable source that demonstrates more hunting in fall, although we know anecdotally this to be the case.  We welcome any suggestions of a citable source by the reviewer for this to be added in the paper. As explained in Ashjian et al (2010), and referenced in the new manuscript, fall hunting seems to be changing as the climate changes due to possible changes in the Pacific Water inflow into the Chukchi Sea (and along with it the euphasesiids the whales consume), as well as hunters are reporting they need to travel further from shore to harvest the whales because the whales are being deflected by increasing vessel traffic.  This causes problems because the meat can spoil on the long tow back to shore.  This as well has to do with the open water period extending into the fall storm season, and the northward shifting storm track, increasing in recent years the number of days reported to be to windy for hunting (see previous Figure).  This discussion has been added to the manuscript in the new subsection titled "Increases in the number of windy days over open water and open water duration."

Pg 6, line 24-27: These statements are not well-supported and may be inaccurate. Does Clarke's paper reference changes in hunting? I suspect not. The bowheads for the BCB stock begin to arrive in the Beaufort in late April/early May, not August. (Perhaps the authors are trying to say that bowheads migrating west from the eastern Beaufort are arriving to the western Beaufort near Pt. Barrow earlier in Fall?) Also, I am not sure there is any literature that shows that the bowheads passage through Bering Strait is tied to local ice conditions there. If there is, it should be referenced. This statement seems quite speculative.

The statement referencing Clarke's paper was unclear. We did not mean for it to reference changes in hunting, but changes in the pattern of the Bowhead whale migration.  We agree that the statement is speculative and it was meant to be worded that way with the words "could be".  However we can see how this might be unclear.  Based also on the reviewers comment below that too much focus is on changes in marine mammals, and also from the limited migration studies we have found (we are open to suggestion by the reviewer), we have decided to remove these statements.

Section 4.3: This entire section that discusses impacts on marine mammals, which rely on large regions and migrate through diverse ice conditions, seems disconnected from the results of this paper, which focus on local conditions near specific communities.

We have greatly shortened this section and the remaining content is now interspersed in the appropriate subsections. However, we still feel it is important to discuss the impacts of sea ice change on marine mammals due to their value for subsistence hunt. For example, in the new subsection of the revised manuscript discussing the increased number of false freeze-ups and false break-ups, we mention this could result in less time or a reduced snow cover on the ice. The latter is important because seals use snow cover for protection from predators. In addition, an earlier break-up (shown in the Figure referenced in the Results section of our revised manuscript pertaining to freeze-up/break-up timing), could lead to problems for the bearded seal, which require stable ice cover in late spring for raising pups and moulting (Kovacs et al, 2011).

Pg 8, Line 8: The authors point out that their simple definition of transition seasons based on sea ice concentration thresholds may be problematic. I agree, and suggest that the authors think carefully about what new data and evidence they can introduce to this paper to make their quantitative results better provide a relevant context for the discussion in this paper on impacts to communities.

We have address this with the development of the new index: the number of false freeze-ups and false break-ups during the seasonal transitions between ice-covered and open water states, instead of the duration of each transition period. We have also added the reference of Serreze et al (2016) to justify our 30% sea ice concentration threshold. Please refer also to our response to Reviewer #1, Data and Methods section, and Reviewer #2, Supplementary comment #2.

Section 5.1: A great explanation of the methodology to recreate the BSI from the HSIA data should be included in the methodology section.

In view of the the comments of the reviewers, we have decided to remove the BSI from the revised version. Please also see Reviewer #1, Discussion comment.

Figures 7 & 8: Why are the upper limits of the BSI shown in Figure 7 not represented in Figure 8 (e.g., values above 1000)? Does Figure 8 correspond to a subset of earlier years?

The BSI has been removed. The two figures have different scales because Figure 8 was meant to be a comparison between our HSIA dataset and the BSI calculated from the other sources. Since we had extended the dataset further than the other sources, we did not include those years in comparison. So yes, in essence, Figure 8 does correspond to a subset of earlier years.

Pg 8, line 30: change to "are very likely"

Yes, corrected.

Pg 9, line 20: Is there any evidence that increased shipping is leading to more goods, and a greater diversity of goods, to Arctic communities?

This statement has been changed so that it is more speculative. However, we did mention in this section that there has been an increase in maritime transit and an associated new recommended shipping route released by the US Coast Guard. We have modified the text to say that an increase in shipping of goods delivery could impact reliance on subsistence foods, especially in young people. While this speculation is only anecdotally supported, we believe it is still worth mentioning here.

We have also added the reference of the Arctic Marine Shipping Assessment (2006) which suggests increased shipping could enhance trade and reduce costs for Arctic communities, and increased development of resources can provide employment and income for Arctic residents.

Pg 10, line 20: This conclusion regarding the traditional spring hunt being cut in half due to ice conditions is not accurate and is not well supported by the data presented in this paper. See my earlier comments. This conclusion, above all else in this paper, should not be published.
We appreciate the reviewers comments here, and have removed (as mentioned in the previous comment about the whaling Discussion section) the comments about any concrete number in the reduction of days left for whaling. We meant (and agree this was not adequately clear) that the trends seen of an earlier break-up leave less time in spring to hunt from the ice. We have now focused on the fall whaling season, where the open water season is expanding further into the fall storm season. This has consequences for the ability for whalers to safely travel by boat and impacts the number of whale landings (Ashjian et al, 2010).

Pg 10, line 24: There is absolutely no evidence presented in this paper or relevant literature cited that indicates a change in bowhead whale migrations.
As mentioned in a previous comment, the focus on whale migration has been removed from the manuscript.

Pg 11, line 5: Where is the evidence that hunters are evaluating risk differently than in the past? The claim that hunters today walk on thinner ice than they used to because of the pressures of environmental change and hunting regulations seems over-simplified and perhaps altogether inaccurate.
The source cited for this statement about hunters is Ford et al (2006), which discusses changes in exposure-sensitivity under a changing climate. However, because the revised manuscript has changed, this statement does not flow well with the discussion of the added indices, so it has been removed.

We appreciate Reviewer #3's comments, which have contributed substantially to a more developed and clearer manuscript.

---

## Author Comment (AC4) · 21 Dec 2017

**Reply to Short Comment #1**

**Impacts of a lengthening open water season on Alaskan coastal communities**

**Rolph *et al.* (2017), tc-2017-211**

We appreciate the input by the two students from the class.  We present below our detailed responses to the comments in red.

**Summary of the manuscript**
The authors have used the Historical Sea Ice Atlas (HSIA) to calculate a date for the break-up and the freeze-up of the sea ice for four coastal Alaska communities (Barrow, Kotzebue, Shishmaref, Nome) as well as for an area in the Bering Strait. The dates were calculated from 1953 to 2013 based on a threshold of 30% ice-cover. Based on this data a linear trend was derived to find a (possibly climate change associated) change in the timing of both freeze-up and break-up. Following this analysis, the paper reviews numerous potential interactions (direct as well as indirect) between the change in sea ice and the impacts on indigenous peoples.

**Main Assessments**
The study discusses a current topic related to climate change, namely the duration of sea ice cover. However, it remains unclear how these communities were selected and why the data for not all communities that were selected (p. 2, l. 30) are presented (figure 2 and 3) and discussed (results, discussion). Additionally, the methods are lacking in detail and statistical details are not addressed. A significant flaw is the lack of an evaluation of the trend line. As the trend line is the main result of this study, it requires an in-depth evaluation and a discussion that compares these results and this method to other studies on changes in sea ice cover. The discussion subsequently doesn't focus on the derived information from the HSIA (the trend line) but more on potential implications of the found changes for the people in those Alaska communities. These implications are based on a literature review, which makes the manuscript two sided and overcharged in information variety. Further the BSI is introduced too late and only covers a short part of the study which poses the question if it is really needed or useful. In summary, the paper in its current state is unfocused and lacks detail in key sections.

The intended focus of this study is the development of climate-related indices from large-scale datasets using local and indigenous knowledge that directly relate to impacts on Alaska coastal communities. We acknowledge that we may have placed undue prominence on the trend lines of open water duration without the appropriate level of discussion later in the manuscript, as the reviewers suggest.   In the revised manuscript, we introduce additional data products (WRF-downscaled ERA-interim reanalysis fields) and develop additional indices that capture aspects of recent sea ice changes relevant to coastal communities.  This allows us to expand our discussion of the observed trends and improve the overall balance of the text by more effectively linking our discussion of community impacts with the results of our analysis.   Please also see our responses to the main comments of the other Reviewers #1, 2, and 3 in response to the students comments about the BSI index.

**General Questions.**

Does the paper present novel concepts, ideas, tools, or data?

Yes, using the HSIA a historical record for the break-up and freeze-up of sea ice for four Alaska communities is presented. However, the literature review (mainly in the discussion section) does not present new findings.

We thank the reviewers for this insight. In the revised manuscript, we believe that our discussion is more focused on the results of our analysis, which now comprise an expanded set of locally-informed climate indices.

Are substantial conclusions reached?

No

Please refer to our responses to Reviewers #1-3, labeled as Question 3. We have provided new conclusions about the recent decades (1979-2014) of the following indices for three communities: Number of false freeze-ups, number of false break-ups, number of open water days deemed too windy for offshore subsistence hunting, number of wind events capable of performing geomorphological work (erosion) or damage to infrastructure or habitats. From 1953-2013, we have also presented changes in the timing of freeze-up and break-up for the communities of Kotzebue and Shishmaref. In Utqiagvik, there has been an approximate tripling of the number of wind events capable of significant coastline erosion from 1979-2014. We believe that our documentation of changes in the various indices have led to new conclusions about environmental changes of dual relevance to Arctic coastal communities.

Are the scientific methods and assumptions valid and clearly outlined?

Not fully, since the methods section lacks a statistical evaluation and later sections (results and discussion) further highlight methodological concepts that were not introduced nor discussed.

We have outlined how we have used each reference pertaining to the appropriate index, and explicitly how we have used the climate-related thresholds (e.g. sea ice concentration or wind speed) in the development of each index timeseries. Please refer to our responses to Reviewers 1-3.

Are the results sufficient to support the interpretations and conclusions?

Not fully, since the derived trend line is not statistically evaluated.

We have added some significance values to the subsection 3.1 Please also see our responses to the comments of Reviewers 1-3, labeled as Question 5.

Is the description of experiments and calculations sufficiently complete and precise to allow their reproduction by fellow scientists (traceability of results)?

Not at all.

We have now provided further methodological details regarding the datasets used and the indices derived from them. We have also excluded any reference to the BSI, which we acknowledge was not described with appropriate detail.

Do the authors give proper credit to related work and clearly indicate their own new/original contribution?

Yes

Does the title clearly reflect the contents of the paper?

No, the title mainly focuses on one aspect of the article (literature review in discussion). A better title would include the derived estimate for a change in the date of freeze-up and break-up. For example, it could be: 60 years of historical ice cover data reveal a significant shift towards a longer open-water season.

We thank the reviewers for this insight, but we feel their suggested title does not adequately capture the intended thrust of this work, which is the application of large-scale data to assess local impacts of changing sea ice and climate for Arctic coastal communities. Instead, we have appended a subtitle to our title, which now reads "Impacts of a lengthening open water season on Alaskan coastal communities: deriving locally-relevant indices from large-scale datasets and community knowledge"

Does the abstract provide a concise and complete summary?
Yes, although there is potential for improvement.
Please see our response to Reviewer #3, Question 9. We have updated the abstract so that it concisely outlines our study communities, how we have developed each community-relevant index, which datasets were used, as well as the main conclusions from the changes seen in the results of the calculated indices timeseries.

Is the overall presentation well-structured and clear?
Partly; owing to the fact that the article presents a mix of a data analysis with a literature review concerning potential impacts for the local communities. A comparison of the changes in sea ice with actual impacts for the communities or potential impacts for four different communities would have been interesting but the impacts are all discussed in a very general and theoretica/hypothetical way. The discussion section is unnecessarily long and does not focus on the actual work done by the authors. Furthermore, the figures are of low quality.
We have refined the scope of the paper such that we clearly outline very specific indices, each of which we provide a timeseries for the three communities examined. The structure now includes these in subsections which are connected in the Methodology, Results, and Discussion sections. We feel this adds much better flow to the paper. We have restructured the Discussion section such that each sub-section (4.1, 4.2, and 4.3) correspond with their own index in the results section, which we believe improves the organization of the manuscript.

Is the language fluent and precise?
Yes

Are mathematical formulae, symbols, abbreviations, and units correctly defined and used?
Mostly
The manuscript contains no mathematical formulae or symbols and the only units used are days and percentage ice concentration (technically a unitless ratio). However, we have now ensured that all abbreviations such as ERA (European Centre for Medium-Range Weather Forecasts Reanalaysis) and WRF (Weather Research and Forecasting model) and now explained.

Should any parts of the paper (text, formulae, figures, tables) be clarified, reduced, combined, or eliminated?
Yes, to all of the above mentioned sections.
We believe we have addressed all the reviewers comments above.

Are the number and quality of references appropriate?
There are no major references to the methodological part of the paper and almost no references to other studies on sea ice cover. The literature review on the other hand seems to have a good basis of literature.
The Methods section of the paper has been changed substantially. In essence, we have cited a reference for each index we derived a timeseries for.

Is the amount and quality of supplementary material appropriate?
NA

Major review points
Abstract
The term "subsistence hunting" could be introduced once, and then it should be clear that the indigenous people of Alaska rely on the availability of game for their food supply, henceforth it would not be required to always indicate it again (also true for the whole article).
We have added a definition to the Introduction section where the term first appears, to provide a background to those not familiar with communities in the Arctic.

Maybe the section that explains the HSIA is not necessary (defining it here as a historical atlas of sea ice cover would be enough).
We have reduced the description of the HSIA in the abstract, such as the resolution of the dataset. However, we respectfully disagree that paraphrasing the name of the dataset we used in the study is valuable for the abstract in terms of describing the study.

Introduction
P.1, L.14 and 17: If the focus is on food security (rather than the impact on coastal communities in general), why is the first example given related to soil erosion? Soil erosion is likely not the main impact on food security.
The introduction has been reorganized, such that the first statements now read: "When trying to downscale large-scale climate observations, complex inter-connections between communities and the environment can often be overlooked (Huntington et al, 2009).  We therefore recognize the importance of incorporating local knowledge in understanding and quantifying the impacts of such direct changes."

 P.1, L.15: the reader is introduced to the terms "direct" and "indirect impacts". However, for the indirect impacts, the term "globally-induced impacts" is also used. In the conclusion again the term indirect impacts is used. We suggest to use the same term in throughout the manuscript.
We have decided to shift the focus of the paper from the differences between direct and indirect impacts, to the new focus of the development of indices defined by thresholds from local knowledge. The term 'globally-induced impacts' does therefore not make an appearance in the new manuscript.

P.1, L.21: Why not cite peer-reviewed literature?
We are not sure what the reviewers are referring to at this line.  Between lines 20-22, there are 6 different  citations to peer-reviewed literature.

P.2, L.3: It is not quite clear what the term "place-based nature of climate change impacts" refers to. A short description or explanation would help.
Yes, we agree that this statement could have been expanded on to make this more clear.  Please see our response to Reviewer #3 about this statement.

Methods
P.2, L.27: the term "best analog representations" was used. What does this mean? For a better understanding of the HSIA it is crucial to know what is meant with "analogs".
Based on the multiple data sources, analog representations are only used when we need to fill spatial and temporal gaps of the given month.  In other words, analog representations assume no large jumps in the sea ice concentration between the first and the last timestep of the HSIA dataset. This does not take place for the majority of the dataset, but for the gaps between 1953-1973.  Our analysis extends

from 1953-2013. Further clarification has been added in the subsection 2.1 "The Historical Sea Ice Atlas" in the revised manuscript.

For this study four communities and one offshore area were chosen. The manuscript states that the communities were chosen because of their "wide range of sea ice regimes, with varying levels of dependence on subsistence activities [...]". For a better understanding of the communities it would be useful to have a short site description for all of them including the reason why a particular location was chosen and further reference to the map in figure 1. Also, it is not clear why this particular location in the Bearing Strait location was chosen.
The Bering Strait is a relatively small area and the sensitivity of our sea ice concentration does not vary much between the grid cells, as there are only about 3 grid cells spanning this with our spatial resolution of the HSIA dataset. We have added a description of each community we chose to focus on in a separate subsection of the Introduction. Please see also our response to Reviewer #2, Comment #1 in the supplementary section.

P.3, L.3: it would be useful to support this claim with a reference to a date for which satellite data would be available (also add a source for the date).
We do give when satellite data is available in the statements in that same paragraph: "One caveat to including multiple data sources is that there are a different the number of observations that have gone into each time segment of data. For example, the frequency of ship-based of observations was not consistent throughout the record and the number of available observations increased dramatically with the advent of the satellite era. However, with regard to the latter, we do not find evidence for anomalous discontinuity in trends around 1978-79, when the passive microwave satellite data are incorporated into the dataset." However, based on the reviewers comments, we have also added the 1979 date to the statement now where we describe use of the HSIA data roughly doubles the timespan we can evaluate when compared to datasets which only use passive microwave satellite data.

P.3, L.10: first a concern is raised that the data is heterogenous, and then it is only partly explained why and how the HSIA is nevertheless a good source. How did you test if there was an anomalous discontinuity?
We looked for anomalous discontinuities in the dataset by looking for large jumps in the dataset for multiple variables, around the time when the satellite data was introduced. We have also added further description to the analog representation in subsection 3.1 in response the reviewers comments.

P.3, L.16ff: it is not clear how the area was selected, was it done manually? How was the calculation then conducted? Overall this section lacks detail and the results are not reproducible.
The area used to extract the sea ice concentration data was offshore the communities, each covering the same area for the communities which is a 50x75 km box near the coastline. These grid cells are highlighted in Figure 1. We have added a separate subsection 2.3 in the Data/Methods section of the revised manuscript which we feel addresses this comment titled "Selection of grid cells representative of each study area". We have added also in the methodology section 2.4 and 2.5 clear descriptions of how we calculated each index in our revised manuscript, since we have added several since the last version.

P.3, L.24: the reasoning to why the 30% threshold was chosen is not clear, if 15% does the same as 30%, then why chose 30%? What did other people do to evaluate freeze-up and break-up of ice cover from gridded data?
Please see our response to Reviewer #2, supplementary comment #2.

Overall: No information on statistical analyses that were used is given. Figure 2 / 3 hint at the use for linear regression, but that is the only information the reader gets from the article. The low "% variance explained" suggests that these trendlines may not be significant? Did you do a statistical test to determine if the trendlines are significant? Also, why was a linear trendline chosen? Does a stepwise (two-part) regression fit the data better?

A linear trend line was chosen because it is a common way to show rates of change over climatological timescales, especially for sea ice decline. To be consistent with this and to apply trends across multiple locations, we therefore do not think a stepwise regression is appropriate, even though it might 'fit the data' better. The significance of the trend lines is included in sections 3.2 and 3.3 of the new manuscript.

Results
Generally, this section relies heavily on the linear trend, although the linear trend is not mentioned in the methods nor is its quality assessed.

The results section of the revised manuscript contains 3 subsections (3.1, 3.2, 3.3). These sections focus on the results of the timeseries of the indices developed, and we have added statistical values to each of these subsections. Also relevant to this comment by the reviewers is our responses to Reviewer #1 Supplementary comments titled "Results." We have removed the freeze-up and break-up trends for Utqiagvik, based on the discontinuity in the number of physical freeze-up and break-up dates starting from the beginning of the dataset toward more recent years.

It would be good to have a table (similar to Table 1) with all important information and statistical measures (not just the % of explained variance).

Since the main focus of the manuscript is demonstrating the use of indigenous knowledge in conjunction with large-scale datasets, we feel a high focus on statistical values is beyond the scope of this paper. We are also not sure what the reviewers mean by "all important information" and would need clarification on that to address this comment further.

The presentation of the Results is incomplete:
- The studied community "Nome" is only mentioned in the Methods. Is there a reason why it is not presented in the Results and Discussion?

Nome is also presented in the Results of the original manuscript (Section 3.2), put we have decided to remove analysis from this community and focus on the three communities of Utqiagvik, Kotzebue, and Shishmaref. Please see also our response to Reviewer #2, supplementary question #1 titled "Community uses of sea ice".

- In Figure 4 suddenly "Wales" appears, without introducing it before.

Please see our response to Reviewer #2, supplementary question #1 titled "Community uses of sea ice".

- Figure 2 and Figure 3 do not contain the results of "Nome" and "Bering Strait". After the introduction of the four communities and one offshore area in the methods, it is necessary to show all results or at least state why something is not shown / presented.

We are consistent in the new manuscript such that each index (with the exception of Utqiagvik for freeze-up/break-up dates) is calculated for each community. Please see our response to Reviewer #2, supplementary question #1 titled "Community uses of sea ice".

- P.4, L.9-15: This focuses on the explained variance by the linear trend, however it is just a qualitative description and raises more questions than it answers no statistical evaluation!
We feel a qualitative description of how much variance is explained by the linear trend is a meaningful value that should be included in the manuscript, but as mentioned above, we feel a detailed statistical analysis of each index developed is beyond the scope of this paper. We have added significance values to sections 3.2 and 3.3.

- In section 3.2, lines 21-24 are already interpretation and should be moved to the discussion.
We have restructured the Discussion section such that each sub-section (4.1, 4.2, and 4.3) correspond with their own index in the results section, which we believe improves the organization of the manuscript.  Please also see our response to Reviewer #1, supplementary question titled "Results".

- Figures 2 and 3: The data for Kotzebue flips back and forth between two values in the 1960s and early 70s. Is this a data limitation issue?
The dates in question are prior to the passive microwave satellite data, and as explained in the Data/Methods section 2.1 "Historical Sea Ice Atlas data", there is an inconsistency in the frequency of data sources included in this time period.  We believe this is the reason for some of the jumps seen in the timeseries prior 1979.

- Figure 4: top 1%, means that for each period a different threshold is chosen. That makes comparisons of the different periods difficult. Also what are the methods used to determine the number of storms? Add a reference.
This figure has been removed from the revised manuscript, as we feel the index timeseries "Number of geomorphological significant wind events" calculated for each community provides more information in this context.  Nonetheless, in the previous manuscript, the reference is given in the caption of the Figure, as well as the method to calculate the number of storms, where the top 1% is an accepted way to calculate extreme events such as storms.

Discussion
- The quality of the produced data is never questioned nor is it assessed. How robust are the results?
The data is directly derived from observational datasets.  The use of satellite-derived sea ice concentration data is a well documented way to obtain sea ice concentration data.

- P4L26: These are at most potential impacts. Unfortunately, the real impacts are never determined or analysed. So change title to reflect that these are literature based potential impacts.
We do not believe that the well-documented case (e.g. Barnhart et al (2014), Overeem et al (2011)) that the Arctic coastline is more vulnerable to erosion due to sea ice loss should be labeled as a 'potential impact', as the reviewers suggest.  Coastal erosion is a very real threat to Arctic communities, for example the case of Shishmaref twice voting to relocate their entire community due to their quickly eroding coastline.  We expand on the impact of sea ice loss in terms of providing more open water during the fall storm season which allows for more waves to develop.  Section 3.3 "Increasing number of wind events with potential for geomorphological change" in the Results and Section 4.3 "Increasing wind events over open water: Number of geomorphologically-significant wind events and consequences for erosion" discuss this in the revised manuscript.  An example of one of our results is that the number of wind events over open water that are capable of causing significant erosion or damage to infrastructure and habitats (Atkinson, 2005) has roughly tripled for the case of Utqiagvik from 1979-2014.

- P.5, L.5ff: the manuscript refers to Kotzebue Sound which "shows less of a change in freeze-up and breakup trends". This could be because it is surrounded by land on three sides. But what about the community of Nome which is on a similar, but not so distinct, location and shows the same trend in figure 4?

Please see our response to Reviewer #1, supplementary comment "Results".

- P5.L34: Is there any evidence for that? Any data? any references?

There is anecdotal evidence that hunters are increasingly relying on larger boats with outboard engines. However, we have not found any peer-reviewed citations of surveys or something of the sort to cite that statement. This statement also does not appear in our new manuscript.

- P7L21: If this has been reported, add the reference.

Please see our response to Reviewer #3, Question #15.

P8.L3: This statement should probably come much earlier.

Please see our response to Reviewer #2, Supplementary Question #2.

P.8 (section 5.1): the manuscript starts again with an introductory part, then a description of the methods, results and discussion. For a clear structure however, a separation into the correct chapters would be necessary. Within this section the Barnett Severity Index (BSI) is very briefly introduced. Here the reader should get more information about the BSI. What is it? Why is it used? The BSI is only mentioned again in the first paragraph of the conclusions and is thus a minor part within the manuscript. Thus, one should introduce it properly and then also show its importance.

Please see our response to Reviewer #1, supplementary subsection "Discussion".

The discussion has a major focus on the impacts. However, there are other aspects as for example species extinction that could be more dramatic with more open water days. Further the protection of species in danger from extinction could lead to conflicts with traditional hunting.

The reviewers are correct that sea ice loss poses a significant threat to certain ice-dependent species, some of which are also subsistence species for coastal communities. However, we feel that issues related to co-management of endangered or threatened species is beyond the scope of this manuscript.

Conclusions
- P.10.L7-9: We agree with this but what is the relation between this statement and your actual study/analyses? The four different sites are not discussed in detail and it remains unclear how different these communities are or how different the impact of climate change and changes in sea ice has been for these communities.

We have added a subsection in our Introduction section (1.2 "Characterization of communities examined") which discusses our study areas, typical sea ice cycles of each, and sea ice use.

- P.10, L.13: the usefulness of the BSI is described. This statement would be more logically placed together with the rest of the BSI, currently in section 5.1.

The BSI has been removed from the revised manuscript. Please see the response to Reviewer #1, labeled "Discussion"

- So far, the direct impacts were always presented before the indirect impacts. On page 10, line 25ff the order was changed which is misleading. Always keep the same order.

We have decided to shift the focus in the revised version from the difference between indirect and direct impacts to the development of locally-relevant indices pertaining to the lengthening open water period, relevant for stakeholders in coastal Arctic communities. Please see the response to Reviewer #1, overall comments. This statement does not exist in the new manuscript.

- Most of the conclusion is actually a discussion and not a conclusion from the analyses presented in this study.

Please see our response to Reviewer #1, Question 10.

**Minor points**

-Some sentences are hard to read, e.g. P2L3-4, L20-21.

The P2 L 3-4 statement has been reworded. L20-21 is not present in the revised manuscript.

-P.2, L.29: Sea ice concentration = sea ice cover. Introduce this definition already in the introduction.

We have added to the introduction sea ice concentration is the fraction of open water covered by sea ice.

-Figure 1: Create a more useful and visually more attractive map. This map looks like it comes out of a video game of the 80s.

Please see our response to Question # 3, Reviewer #2. The reason we used this map was to highlight the resolution of the HSIA gridded dataset and feel that the map provides more information when presented in this way rather than with a finer resolution we did not use.

-The legends in figure 2, figure 3 are too small and do not contain the necessary information.

The figures with the legends are not present in the revised manuscript.

- Figure 5: don't just copy from Chapman, replot using the same style as the other plots. In general, all plots should have the same style, so also figure 4 needs to be adapted.

This figure is not present in the revised manuscript.

-Figure 8: add the 1:1 line

This figure is not present in the revised manuscript.

-Table 1: maybe it is better to give the trend in days per decade so that it is not a fraction of a day, which could suggest hourly data are needed

We feel that days/year is consistent with the other trends given throughout the manuscript (Results: Section 3), and have decided to keep as is.

-Regarding the figures in general: no titles needed, clearer figure captions needed (a,b,c...). Also make sure to add informative legends.

P.3, L.8: the word "of" is not needed

We have looked over this line and are unsure which 'of' would be able to be taken out.

P.4, L.21: replace "has" with "could have" because we cannot be sure about this.

We are sure that changes in oceanic heat impact influences when the ocean is able to freeze.

P.6, L.25: "polynas" actually called "polynyas"

This is a typo and this statement does not exist in the new manuscript.

P.8, L.12: the introduction of the BSI is suboptimal. It is here introduced as "severity of ice conditions index" whereas the acronym itself translates to "Barnet Severtity Index". Only after the first introduction of the correct terminology abbreviations should be used.

As mentioned previously, the BSI has been removed from the new manuscript.

P.8, L.24: also give the 5 nautical miles in kilometers (in general only use one distance measure)

The BSI has been removed from the new manuscript and so this statement is no longer present in the revised version.

P.9, L.14: "will" or "can"?

This typo has been fixed in the new manuscript.

---

## Referee Report (RR1)

[referee-annotated manuscript omitted]

---

## Author Response (AR2)

Author response to Reviewer #1

We thank Reviewer #1 for the thoughtful and helpful comments. The responses by the authors are written in blue.

The authors thoroughly responded to the first round of reviewer comments. The paper is now much more focused and better utilizes the data presented to draw conclusions. I have attached a pdf with many comments embedded. Most comments are minor. Below, I list the areas that I believe require a bit more attention before the paper is published.

1. The title references "community knowledge" and the first sentence of the abstract states that this study is "Using thresholds of physical climate variables developed directly from indigenous knowledge". I find these claims of using community and indigenous knowledge too generous in describing the approach of this study. There are references to wind thresholds that are dangerous, references to how hunting is impacted by barge noise and ice conditions, etc., but I think it is problematic to claim that this paper used "indigenous knowledge" in the study. This is not a critique of the study itself, but rather in how it is described. This is certainly a paper that is applicable to these communities and uses community "observations" in their methodoloy and discussion, but use of knowledge is a much different thing, especially as we are seeing the raising of the bar for what is classified as indigenous knowledge research and knowledge co-production.
Yes, we agree that our wording was misleading here, especially with the recent increase in the use of indigenous knowledge in Arctic studies. We have changed the title and the quoted sentence of the abstract (as well as some other related sentences in the manuscript) to reflect that the physical climate thresholds used in the study are made from local observations, rather than using the term 'indigenous knowledge'. In instances where the thresholds were obtained from community engagement reported in other published studies, we cite those studies.

2. Figure 1 shows the grid cells used to extract the sea ice concentration and weather data, which provides much of the basis for this paper's analysis. The large flaw lead system that develops off of Alaska's North Slope in winter and spring typically occurs south of the grid cells chosen to represent conditions off Utqiagvik. Therefore, I am concerned that the location for the ice analysis doesn't represent conditions near the village, especially in spring. This should be discussed in the paper, and the selection for grid cells better justified.
We see the need to justify the selection of the grid cells, especially in consideration of flaw leads and coastal polynyas that are common near Utqiagvik, and we thank the reviewer for his/her comment. We have made the subsection in the manuscript describing grid cell selection clearer about why we chose the grid cells. We have added the reviewer's valid point about the flaw lead system to the manuscript as well. Unfortunately, the interpolated ERA-Interim data used for lower boundary forcing is most accurate away from coastlines. This is due to the contamination of the sea ice signal by the land-ocean boundary. Because of this, moving the grid cells southward and closer to the coastline would likely degrade the analysis. We have also added a reference in Section 4.1, p. 9, L16, which states: "Although the grid cells selected for our study do not capture landfast ice at Shishmaref and Utqiaġvik, Mahoney et. al (2014) found trends toward earlier break-up of landfast ice along the Alaska coast in agreement with those identified here."

3. The paper analyzes severe winds events capable of geomorphological change. In this regard, wind direction is of high importance, yet the authors only look at scalar values. For example, onshore and downwelling winds will have more potential for coastal sediment erosion and storm surge. At Utqiagvik, it is stormy winds from the West and NW that are most likely to cause coastal change or

damage. Why have the authors not looked at specific wind components to characterize sever wind events?

We thank the reviewer in particular for this comment, because it contributes to the practical aspects of the manuscript. We have computed the wind directions and the number of hours the winds are between two 'direction thresholds' defining the onshore/alongshore quadrant for each community.  We then presented the number of these severe wind events (which the reviewer mentions above) that are composed of winds between these threshold directions.  We have added the results as a second y-axis in Figure 8,  and explained the method in the caption of that figure, and also in a newly added portion of the methods (subsection 2.5, p. 5 L30).  The wind directions in the alongshore direction generally promote water setup along the coast, and promote water downwelling. Winds from this direction, as the reviewer mentions in his/her comment, have more potential for storm surge and coastal erosion.  The inclusion of the resulting occurrences adds to the analysis of the data we have available by directly linking it to impacts on coastal communities.  If we take Utqiaġvik as an example, we have presented the number of the severe wind events which have winds coming from the directions of Northwest (representing onshore towards Utqiaġvik) and Southwest (representing alongshore winds promoting downwelling).  Kotzebue and Shishmaref, the other two communities examined in the manuscript, have different direction thresholds to calculate the number of the wind events that are between alongshore and alongshore/downwelling, due to their different orientations along the Alaska coastline.  The directions for Kotzebue are between West and South, and for Shishmaref, between NW and SW.  Please see the methods subsection 2.5 for further detail. We have added the results of this analysis in the Results subsection 3.3 (p. 7 L25) , and added discussion of these results in Section 4.3 (starting at p. 11 L8).

4. Lastly, I believe this paper will eventually make a nice contribution to the literature. Of greatest potential value (in my opinion) is the opportunity to illuminate the different ice regimes and conditions across these three communities. The differences are mentioned, but could be discussed a bit more in the conclusions, especially in the sense of what may be expected with further change. The paper makes a lot of general, broad comments on community impacts. It could be improved with a few more comments specific to these three communities.

Thank you for this comment.  The addition of several statements in the conclusion section, comparing the differences in the indices across the communities has, we believe, improved the presentation. We have lengthened the conclusion section in the new version of the manuscript by comparing the indices across the communities and linking them to particular impacts associated with the three different communities.  We have also added a statement that the increase in the wind events coming from the NW and SW in Utqiaġvik (new results added to the manuscript, based on the Reviewer #1's comment # 3 above) is likely to be a problem in the future if this trend were to continue in terms of erosion and the relatively greater amount of infrastructure present in Utqiaġvik.

**The responses below pertain to the detailed comments marked within the manuscript by the reviewer.  The author responses are in blue and the comments by the reviewer are in black.**

**p. 2 , L 23:** Why not identify the location of these three communities in Figure 1?

In response to this suggestion, we have amended the map to include the community names and locations.

**p. 2, L. 29:** should be "sheefish and bearded and ringed seals"

Thanks, corrected

**p. 2 L 34:** Be more specific. They have voted to "permanently relocate the village"

Yes, this has been corrected

**p. 3 L 4:** The village is actually along the Chukchi Sea.
Thanks, this has been corrected.

**p. 3 L 8:** I suggest sticking with one term - either "landfast" or "shorefast"..
Agreed.  We have gone with "landfast".

**p. 3 L 20:** Insert "(HSIA)" immediately after first use.
Thanks, this has been inserted at what is now p. 3 L25.

**p. 4 L 4** You are getting sea ice concentration from this product correct? This though is not explicitly stated in this paragraph.
Yes, and we also obtained the wind data from this product.  Although we did mention the use in the subsequent section, we did not explicitly state it in this section.  We have added such a statement to this subsection (p. 4 L17) and provided further detail about the dataset.

**p. 4 L 16** The large flaw lead system that develops off of Alaska's North Slope typically occurs south of the grid cells chosen to represent conditions off Utqiagvik. Therefore, I am concerned that the location for the ice analysis doesn't represent conditions near the village, especially in Spring.
This is an important point and we have added the description of the flaw lead in this subsection, because the reader should definitely be made aware of this flaw lead during Spring.  The HSIA sea ice concentration data and WRF-downscaled ERA interim (used in this study) rely heavily on satellite-derived sea ice concentration products, which are problematic very close to the shoreline.  Please see our above response #2 in the section "main comments".  Due to limitations in the satellites' ability to distinguish land from frozen ocean near coastlines, moving the grid cells southward and closer to the coastline would reduce the accuracy of the concentrations.  We have also added a reference (p. 9, L16) which mentions that the trends in landfast ice agree with our results presented in the manuscript. We have added a statement about the flaw lead in section 2.3, p. 4, L 27.

**p. 4 L 18** "Security" is a vague term to use here. Why not just say "relevant to each community"?
Yes, agreed. We have changed the statement so that it now reads "To assess variability of ice conditions, we selected data near the coastline of each community but not too close due to the fact a large part of the sea ice dataset is satellite-derived." (Section 2.3, L23)

**p. 5 L 5** Shouldn't these sentences which describe why the HSIA uses a higher threshold be immediately after the first sentence in this paragraph, which mentions the two different thresholds. Still, however, I am not clear how a higher threshold of 30% provides a "quantitative value for comparison between communities". Can't 15% also be the basis for comparison?
Yes, 15% can also be a basis for comparison. The point we were trying to make was that as long as a threshold is selected, the same threshold can be used across communities, thereby providing a basis for comparison.  The statements here now read: "We experimented with different concentration values and found that the resulting dates of freeze-up and break-up are relatively insensitive to thresholds between 15% and 30%. To illustrate this, we used a threshold of 30% for the HSIA data, as used by Serreze et al. (2016), and a threshold of 15% as used by NSIDC, for the ERA-Interim data (Figure 2)."
          Figure 2 is a new figure added to the manuscript in response to this Reviewer comment (see response to next reviewer comment for p.5 L10).  We have tried to make it clearer in the text that having a threshold provides a useful basis for comparison between communities, but the results of this comparison are relatively insensitive to the choice of threshold, since different thresholds produce

similar results in number of open water days. We have also added in the conclusion section a statement that if enough community observations were available for freeze-up and break-up timing, then such a threshold may not be necessary.

**p. 5 L 10** Instead of comparing HSIA at 30% and ERA-Interim at 15% to show that there is not a large difference in the number of open water days, why not compare ERA-Interim at 30% and ERA-Interim at 15%? That would make more sense scientifically. That comparison would provide a nice plot for the paper as well. The same could be included for comparing HSIA at 30% to HSIA at 15% concentration. This comment indicates to the authors that we should have been more clear in the wording here, and we thank the reviewer for this comment. We have rearranged the text and added some clarifying statements in this section. Since the HSIA dataset does not contain any wind data, we have to use the WRF-downscaled ERA Interim wind and sea ice data to help evaluate the days which are 'too windy' for hunting and to identify the number of 'geomorphologically significant' wind erosion events. These events occur over open water, so the sea ice concentration had to also be utilized from the ERA dataset, in order to identify the high-wind occurrences over open water. Also in response to this comment, we have added a new figure (Figure 2) to support the statement that the 15% and 30% thresholds result in very similar numbers of open water days. We feel that a more extensive comparison of how open water changes across the two datasets with the different thresholds is a good idea, but also that a thorough evaluation of the sensitivity of the number of open water days to sea ice concentration thresholds across the two datasets is beyond the scope of the study.

**p. 5 L 13** This is a good point, but these last two sentences should be included in the discussion, alongside some discussion of the challenges. Its important to recognize the challenges of getting communities to report freeze-up since there are many different ways that locals may define freeze-up. Thanks, we agree that this statement is better moved somewhere else. We have moved it to the conclusions section, because we think that further exploration into how freeze-up is defined is also an important topic for future research. The statements regarding this topic in the conclusion section (p. 12 L 25) now read: "Indices such as those we have derived here show that the use of community observations and local knowledge in conjunction with large-scale climate datasets can be a powerful tool in evaluating the impacts of climate change at local scales. All of our results relied on the use of a sea ice concentration threshold to identify transitions between periods of open water or ice cover. For further research, it would be useful for individual communities to report when freeze-up and break-up occur. If every community provided this information, choosing a threshold for sea ice concentration may not be required in comparative analyses of large-scale datasets across communities."

**p. 5 L 22** Correct to "refers to the number of days that sea ice concentration is below..."
Thanks, this has been corrected.

**p. 6 L 4** Correct to: "..prior to the starting date of..."
Yes, thanks, corrected.

**p. 6 L6** Figures 3 and 4 are referenced before Figure 2. Should the order be corrected?
Figures have been re-ordered, thanks.

**p. 6 L20** This is interesting analysis. It would be valuable to know the weeks of year where these false-events took place, either in a plot or table.
We thank the reviewer in particular for this comment because it allowed us to better separate the time period of open water and to better distinguish between the false freeze-ups and false break-ups. We have also added a supplementary table to the manuscript which gives the month and day of each false

break up and false freeze-up, for each community. We have updated some of those results in Subsection 3.1 (p. 6, L 21) and Figure 5, based on the dates of the false freezeup/breakups. The supplementary table is mentioned in Subsection 3.1 (p.6 L28).

**p. 6 L29** This is a long section title, and includes a conclusion. This is inconsistent with the other section titles in the paper. And having a conclusion in a results section title is a bit odd.
The title of the section has been renamed to "Changes in the number of days 'too windy' for safely hunting via boat"

**p. 6 L29** See comment at Figure 5 below. . Comment at Figure 5: 'Please clarify why the two parallel bars for each five year period are not equal in height. Is it because the blue is the HSIA data at 30% and the red/green is based on the ERA-interim at the15% threshold? It could be more informative to just show the red/green bars for each year (1979-2014), and leave off the blue. '
We had previously not included the HSIA-derived results (blue bars) but in the end decided to include them in order to show the longer timeseries alongside the ERA-Interim.  However, due to the differences in the sea ice data incorporated into the HSIA and ERA-Interim, and sea ice concentration threshold, as well as the different grid cell area, we now see that the reviewer's comment makes sense, so we have omitted the blue bars. This allows the reader to focus on the main point being conveyed in this figure: the comparison between the total number of open water days and the number of boatable days for each community.  The blue bars (HSIA) are omitted in the edited Figure 7 in the most recent manuscript version (previously Figure 5).

**p. 7 L9** The results reported appear correct, but it is a bit confusing/misleading. Increases in open water days in part drive the number of days that are too windy because there is a greater potential for days that are too windy with more open water days. Isn't the number of total boatable days (not expressed in relative terms) what is ultimately important?
The changes were presented in relative terms to show that just because there is a rate of increase in the number of open water days, there is not necessarily the same rate of increase in number of boatable days (due to high wind over open water).  However, we do agree that the number of boatable days is ultimately what is important, so,  based on this reviewer comment, we have removed the statements explaining the relative percentage changes.  We have also added a few statements in this subsection about the rates of changes in the total number of boatable days between the communities. These changes are in section 3.2, p. 7, L4 and section 4.2, p. 10, L23.

**p. 7 L10** Note that you are using a mix of both 6-year and 5-year bins in the figure. Clarifying why or finding a more consistent approach is needed.
Thanks for pointing out the potential confusion.  We have changed Figure 7 so that each bin now includes 4 years of data. This results in a total of 9 bins for our 36-year timeseries.  As suggested by the reviewer in a previous comment, we have also omitted the blue bars (HSIA) from this figure.

**p. 7 L15** Wind direction is ultimately of high importance to the potential for geomorphological change. For example, onshore and downwelling winds will have more potential for coastal sediment erosion and storm surge. (At Barrow, it is stormy winds from the West and NW that are most likely to cause coastal change or damage.) Why not look at specific wind components instead of the scalar value?
This is a very good point.  In response, we have augmented this section to include the relevant information on wind direction. We have added a second y-axis to Figure 8 in order to capture the number of those high wind events which are onshore and alongshore (downwelling).  We have added a description of the procedure for identifying onshore/downwelling high-wind events in the Methods

subsection 2.5 (p. 5 L 30). Please also see our response to the Reviewer comment above, which is related to this one (Reviewer #1 main comment #3).

**p. 7 L32** Again, it is not clear why the relative number of high wind events to the total number of open water days is important. Isn't the TOTAL number of high wind events (during periods of open water) what is ultimately important, especially considering that this paper is looking at local community impacts?

The authors somewhat address the value of the relative value in the discussion (windy days of open water versus being able to travel by snowmachine on the ice) but it is not clear here.
We now state the results in terms of the absolute wind events.

**p. 8 L13** It seems that the underlying premise here is linked to uncertainty. With an expanding transition season, characterized by variability, the communities may have a harder time deciding on when and how to change their modes of transportation and hunting. If the authors agree, I think seasonal uncertainty is an important point to elaborate on.
Yes, this is exactly the point we were trying to get across in this subsection.  In response to the reviewer's comment, we have added the following statements to this section:
p. 8 L7: "Additionally, the twice-annual transitions between open-water and ice-covered seasons are becoming increasingly ill-defined and characterized by multiple "false" freeze-up and break-up events before the final, lasting transition occurs (Figure 5)."
p. 8 L 11:  "Once the ocean surrounding the community starts freeze-up, transportation via small boats becomes increasingly difficult and risky.  Hence, until a stable landfast sea ice cover forms (allowing the use of snow machines, dog-sleds, or regular street vehicles in some cases) early winter travel to the mainland from villages on islands and peninsulas such as Shishmaref and Kotzebue can be extremely limited.  The growing number of false freeze-up events each year extends this period of reduced accessibility and increases the level of uncertainty about land at this time of year."

**p. 8 L18** It is not clear why the number of false freeze-ups has implications for coastal erosion rates. There may be a realationship but it is not well expressed here. Isn't it just the presence of open water (ice concentration) that is ultimately important? For example, a year with many false freeze-ups could have more ice in a region over the fall period compared to a year with very few false freeze-ups but with a very late date of freeze-up.
We have added a more detailed explanation in this section. We think this addition contributes to the discussion of why the recent increase in number of false freeze-ups is important for erosion.  We have also added a reference here (Eicken et al., 2005) to explain that the increase in the number of false freeze-ups could increase sediment load transport (Section 4.1, p. 8, L 33).

**p. 8 L26** Correct to: "...false break-ups at Utqiagvik…"
This statement now refers to all three communities.

**p. 8 L27** "turbulent" is an very imprecise word choice. I suggest "more variable".
This section has been revised and the statement containing "turbulence" no longer exists.

**p. 8 L 29** It is not clear what "uneven" refers to. Is there a more precise word?
"Uneven ice deterioration" here was referring to the spatial variability in sea ice melt.  We have changed the statements here to refer to the reduced accessibility of sea ice travel (section 4.1, p. 8 L30)

**p. 11 L26.** replace 'in' with 'near'
Replaced, but this statement is now found at p. 11 L7.

**p. 12 L04.** Remove 'to'
Thanks, removed.

**p. 12 L06.** Remove 'previously Barrow'
Removed.

Author response to Reviewer #2

Author response is written in green, and Reviewer #2 comment is written in black.

While somewhat improved over the previous version this manuscript still suffers from a lack of clarity and concision in its description of the objectives, explanations of analyses and results, and poorly-proofed writing. The manuscript is still so far from being in at least in 'near-final' condition that I cannot recommend publication.

We thank the reviewer for his/her perspective, and we have made extensive editorial revisions to the manuscript with this in mind. We have enlisted a former editor for assistance in revising the presentation for clarity and technical correctness. The description of the objectives mentioned above by the reviewer is provided at the start of the manuscript: "Using thresholds of physical climate variables developed directly from community observations, together with two large-scale datasets, we have produced local indices directly relevant to the impacts of a reduced sea ice cover on Alaska coastal communities." To address the reviewer's comment about the "explanation of the analyses and results", we note that in response to the comments of the other reviewer, the results of the indices are now given in separate subsections, which are ordered (numbered) consistently throughout the methods, results, and discussion sections. We believe that the organization of the manuscript has been improved from the initial submission by switching to this section/subsection structure. Concerning the "poorly-proofed" writing, we have proof-read the text, double checked all Figure references, and revised the wording based on the specific comments and line numbers given by Reviewer #1 (see responses to Reviewer 1).

Summary
4/21/2018 1:33:07 PM

Differences exist between documents.

**New Document:**
cryosphere_submission_v3
26 pages (5.73 MB)
4/21/2018 1:32:53 PM
Used to display results.

**Old Document:**
cryosphere_submission_v2
25 pages (5.79 MB)
4/21/2018 1:32:53 PM

Get started: first change is on page 1.

No pages were deleted

**How to read this report**

**Highlight** indicates a change.
 indicates deleted content.
▲ indicates pages were changed.
⬌ indicates pages were moved.

[revised manuscript text omitted]